# FINITE DEPTH AND WIDTH CORRECTIONS TO THE NEURAL TANGENT KERNEL

**Boris Hanin**
Department of Mathematics
Texas A&M University
College Station, TX 77843, USA
bhanin@math.tamu.edu

**Mihai Nica**
Department of Mathematics
University of Toronto
Toronto, Canada
mnica@math.utoronto.ca

## ABSTRACT

We prove the precise scaling, at finite depth and width, for the mean and variance of the neural tangent kernel (NTK) in a randomly initialized ReLU network. The standard deviation is exponential in the ratio of network depth to width. Thus, even in the limit of infinite overparameterization, the NTK is not deterministic if depth and width simultaneously tend to infinity. Moreover, we prove that for such deep and wide networks, the NTK has a non-trivial evolution during training by showing that the mean of its first SGD update is also exponential in the ratio of network depth to width. This is sharp contrast to the regime where depth is fixed and network width is very large. Our results suggest that, unlike relatively shallow and wide networks, deep and wide ReLU networks are capable of learning data-dependent features even in the so-called lazy training regime.

## 1 INTRODUCTION

Modern neural networks are typically overparameterized: they have many more parameters than the size of the datasets on which they are trained. That some setting of parameters in such networks can interpolate the data is therefore not surprising. But it is *a priori* unexpected that not only can such interpolating parameter values can be found by stochastic gradient descent (SGD) on the highly non-convex empirical risk but also that the resulting network function generalizes to unseen data. In an overparameterized neural network $\boldsymbol{N}(x)$ the individual parameters can be difficult to interpret, and one way to understand training is to rewrite the SGD updates

$$\Delta\theta_p \;=\; -\lambda\,\frac{\partial \mathcal{L}}{\partial \theta_p}, \qquad p = 1, \ldots, P$$

of trainable parameters $\theta = \{\theta_p\}_{p=1}^P$ with a loss $\mathcal{L}$ and learning rate $\lambda$ as kernel gradient descent updates for the values $\boldsymbol{N}(x)$ of the function computed by the network:

$$\Delta\boldsymbol{N}(x) \;=\; -\lambda\,\langle K_{\boldsymbol{N}}(x,\cdot), \nabla\mathcal{L}(\cdot)\rangle \;=\; -\frac{\lambda}{|\mathcal{B}|}\sum_{j=1}^{|\mathcal{B}|} K_{\boldsymbol{N}}(x, x_j)\frac{\partial \mathcal{L}}{\partial \boldsymbol{N}}(x_j, y_j). \tag{1}$$

Here $\mathcal{B} = \{(x_1, y_1), \ldots, (x_{|\mathcal{B}|}, y_{|\mathcal{B}|})\}$ is the current batch, the inner product is the empirical $\ell_2$ inner product over $\mathcal{B}$, and $K_{\boldsymbol{N}}$ is the *neural tangent kernel* (NTK):

$$K_{\boldsymbol{N}}(x, x') \;=\; \sum_{p=1}^{P}\frac{\partial \boldsymbol{N}}{\partial \theta_p}(x)\frac{\partial \boldsymbol{N}}{\partial \theta_p}(x').$$

Relation (1) is valid to first order in $\lambda$. It translates between two ways of thinking about the difficulty of neural network optimization:

(i) The parameter space view where the loss $\mathcal{L}$, a complicated function of $\theta \in \mathbb{R}^{\#\text{parameters}}$, is minimized using gradient descent with respect to a simple (Euclidean) metric;

(ii) The function space view where the loss $\mathcal{L}$, which is a simple function of the network mapping $x \mapsto \boldsymbol{N}(x)$, is minimized over the manifold $\boldsymbol{M_N}$ of all functions representable by the architecture of $\boldsymbol{N}$ using gradient descent with respect to a potentially complicated Riemannian metric $K_{\boldsymbol{N}}$ on $\boldsymbol{M_N}$.

A remarkable observation of Jacot et al. (2018) is that $K_{\boldsymbol{N}}$ simplifies dramatically when the network depth $d$ is fixed and its width $n$ tends to infinity. In this setting, by the universal approximation theorem (Cybenko, 1989; Hornik et al., 1989), the manifold $\boldsymbol{M_N}$ fills out any (reasonable) ambient linear space of functions. The results in Jacot et al. (2018) then show that the kernel $K_{\boldsymbol{N}}$ in this limit is frozen throughout training to the infinite width limit of its average $\mathbb{E}[K_{\boldsymbol{N}}]$ at initialization, which depends on the depth and non-linearity of $\boldsymbol{N}$ but not on the dataset.

This mapping between parameter space SGD and kernel gradient descent for a fixed kernel can be viewed as two separate statements. First, at initialization, the distribution of $K_{\boldsymbol{N}}$ converges in the infinite width limit to the delta function on the infinite width limit of its mean $\mathbb{E}[K_{\boldsymbol{N}}]$. Second, the infinite width limit of SGD dynamics in function space is kernel gradient descent for this limiting mean kernel for any fixed number of SGD iterations. As long as the loss $\mathcal{L}$ is well-behaved with respect to the network outputs $\boldsymbol{N}(x)$ and $\mathbb{E}[K_{\boldsymbol{N}}]$ is non-degenerate in the subspace of function space given by values on inputs from the dataset, SGD for infinitely wide networks will converge with probability 1 to a minimum of the loss. Further, kernel method-based theorems show that even in this infinitely overparameterized regime neural networks will have non-vacuous guarantees on generalization (Wei et al., 2018). However, as (Wei et al., 2018) shows, the regularized neural networks at finite width can have better sample complexity the corresponding infinite width kernel method.

But replacing neural network training by gradient descent for a fixed kernel in function space is also not completely satisfactory for several reasons. First, it suggests that no feature learning occurs during training for infinitely wide networks in the sense that the kernel $\mathbb{E}[K_{\boldsymbol{N}}]$ (and hence its associated feature map) is data-independent. In fact, empirically, networks with finite but large width trained with initially large learning rates often outperform NTK predictions at infinite width (Arora et al., 2019). One interpretation is that, at finite width, $K_{\boldsymbol{N}}$ evolves through training, learning data-dependent features not captured by the infinite width limit of its mean at initialization. In part for such reasons, it is important to study both empirically and theoretically finite width corrections to $K_{\boldsymbol{N}}$. Another interpretation is that the specific NTK scaling of weights at initialization (Chizat & Bach, 2018b;a; Mei et al., 2019; 2018; Rotskoff & Vanden-Eijnden, 2018a;b) and the implicit small learning rate limit (Li et al., 2019) obscure important aspects of SGD dynamics. Second, even in the infinite width limit, although $K_{\boldsymbol{N}}$ is deterministic, it has no simple analytical formula for deep networks, since it is defined via a layer by layer recursion. In particular, the exact dependence, even in the infinite width limit, of $K_{\boldsymbol{N}}$ on network depth is not well understood.

Moreover, the joint statistical effects of depth and width on $K_{\boldsymbol{N}}$ in *finite size networks* remain unclear, and the purpose of this article is to shed light on the simultaneous effects of depth and width on $K_{\boldsymbol{N}}$ for finite but large widths $n$ and any depth $d$. Our results apply to fully connected ReLU networks at initialization for which our main contributions are:

1. In contrast to the regime in which the depth $d$ is fixed but the width $n$ is large, $K_{\boldsymbol{N}}$ is *not* approximately deterministic at initialization so long as $d/n$ is bounded away from $0$. Specifically, for a fixed input $x$ the normalized on-diagonal second moment of $K_{\boldsymbol{N}}$ satisfies

$$\frac{\mathbb{E}\left[K_{\boldsymbol{N}}(x,x)^2\right]}{\mathbb{E}\left[K_{\boldsymbol{N}}(x,x)\right]^2} \simeq \exp(5d/n)\left(1 + O(d/n^2)\right).$$

Thus, when $d/n$ is bounded away from $0$, even when both $n, d$ are large, the standard deviation of $K_{\boldsymbol{N}}(x,x)$ is at least as large as its mean, showing that its distribution at initialization is not close to a delta function. See Theorem 1.

2. Moreover, when $\mathcal{L}$ is the square loss, the average of the SGD update $\Delta K_{\boldsymbol{N}}(x,x)$ to $K_{\boldsymbol{N}}(x,x)$ from a batch of size one containing $x$ satisfies

$$\frac{\mathbb{E}\left[\Delta K_{\boldsymbol{N}}(x,x)\right]}{\mathbb{E}\left[K_{\boldsymbol{N}}(x,x)\right]} \simeq \frac{d^2}{nn_0}\exp(5d/n)\left(1 + O(d/n^2)\right),$$

where $n_0$ is the input dimension. Therefore, if $d^2/nn_0 > 0$, the NTK will have the potential to evolve in a data-dependent way. Moreover, if $n_0$ is comparable to $n$ and $d/n > 0$ then it is possible that this evolution will have a well-defined expansion in $d/n$. See Theorem 2.

In both statements above, $\simeq$ means is bounded above and below by universal constants. We emphasize that our results hold at finite $d, n$ and the implicit constants in both $\simeq$ and in the error terms $O(d/n^2)$ are independent of $d, n$. Moreover, our precise results, stated in §2 below, hold for networks with variable layer widths. We have denoted network width by $n$ only for the sake of exposition. The appropriate generalization of $d/n$ to networks with varying layer widths is the parameter

$$\beta := \sum_{i=1}^{d} \frac{1}{n_j},$$

which in light of the estimates in (1) and (2) plays the role of an inverse temperature.

## 1.1 PRIOR WORK

A number of articles (Bietti & Mairal, 2019; Dyer & Gur-Ari, 2019; Lee et al., 2019; Yang, 2019) have followed up on the original NTK work Jacot et al. (2018). Related in spirit to our results is the work Dyer & Gur-Ari (2019), which uses Feynman diagrams to study finite width corrections to general correlations functions (and in particular the NTK). The most complete results obtained by Dyer & Gur-Ari (2019) are for deep linear networks but a number of estimates hold general non-linear networks as well. The results there, like in essentially all previous work, fix the depth $d$ and let the layer widths $n$ tend to infinity. In contrast, our results (as well as those of Hanin (2018); Hanin & Nica (2018); Hanin & Rolnick (2018)), do not treat $d$ as a constant, suggesting that the $1/n$ expansions (e.g. in Dyer & Gur-Ari (2019)) can be promoted to $d/n$ expansions. Also, the sum-over-path approach to studying correlation functions in randomly initialized ReLU nets was previously taken up for the forward pass by Hanin & Rolnick (2018) and for the backward pass by Hanin (2018) and Hanin & Nica (2018). We also point the reader to Theorems 3.1 and 3.2 in Arora et al. (2019), which provide quantitative rates of convergence for both the neural tangent kernel and the resulting full optimization trajectory of neural networks at large but finite width (and fixed depth).

## 2 FORMAL STATEMENT OF RESULTS

Consider a ReLU network $N$ with input dimension $n_0$, hidden layer widths $n_1, \ldots, n_{d-1}$, and output dimension $n_d = 1$. We will assume that the output layer of $N$ is linear and initialize the biases in $N$ to zero. Therefore, for any input $x \in \mathbb{R}^{n_0}$, the network $N$ computes $N(x) = x^{(d)}$ given by

$$x^{(0)} = x, \quad y^{(i)} := \widehat{W}^{(i)} x^{(i-1)}, \quad x^{(i)} := \text{ReLU}(y^{(i)}), \qquad i = 1, \ldots, d, \tag{2}$$

where for $i = 1, \ldots, d-1$

$$\widehat{W}^{(d)} := (1/n_{i-1})^{-1/2} W^{(i)}, \quad \widehat{W}^{(i)} := (2/n_{i-1})^{-1/2} W^{(i)}, \qquad W^{(i)}_{\alpha, \beta} \sim \mu \ i.i.d., \tag{3}$$

and $\mu$ is a fixed probability measure on $\mathbb{R}$ that we assume has a density with respect to Lebesgue measure and satisfies:

$$\mu \text{ is symmetric around } 0, \qquad \text{Var}[\mu] = 1, \qquad \int_{-\infty}^{\infty} x^4 d\mu(x) = \mu_4 < \infty. \tag{4}$$

The three assumptions in (4) hold for virtually all standard network initialization schemes with the exception of orthogonal weight initialization. But believe our results extend hold also for this case but not do take up this issue. The on-diagonal NTK is

$$K_N(x, x) := \sum_{j=1}^{d} \sum_{\alpha=1}^{n_{j-1}} \sum_{\beta=1}^{n_j} \left( \frac{\partial N}{\partial W^{(j)}_{\alpha, \beta}}(x) \right)^2 + \sum_{j=1}^{d} \sum_{\beta=1}^{n_j} \left( \frac{\partial N}{\partial b^{(j)}_{\beta}}(x) \right)^2, \tag{5}$$

and we emphasize that although we have initialized the biases to zero, they are not removed from the list of trainable parameters. Our first result is the following:

**Theorem 1** (Mean and Variance of NKT on Diagonal at Init). *We have*

$$\mathbb{E}[K_{\boldsymbol{N}}(x,x)] = d\left(\frac{1}{2} + \frac{\|x\|_2^2}{n_0}\right).$$

*Moreover, we have that $\mathbb{E}[K_{\boldsymbol{N}}(x,x)^2]$ is bounded above and below by universal constants times*

$$\exp\left(5\beta\right)\left(\frac{d^2\,\|x\|_2^4}{n_0^2} + \frac{d\,\|x\|_2^2}{n_0}\sum_{j=1}^{d}e^{-5\sum_{i=1}^{j}\frac{1}{n_i}} + \sum_{\substack{i,j=1\\i\leq j}}^{d}e^{-5\sum_{i=1}^{j}\frac{1}{n_i}}\right), \qquad \beta = \sum_{i=1}^{d}\frac{1}{n_i}$$

*times a multiplicative error $\left(1 + O\left(\sum_{i=1}^{d}\frac{1}{n_i^2}\right)\right)$. In particular, if all the hidden layer widths are equal (i.e. $n_i = n$, for $i = 1, \ldots, d-1$), we have*

$$\frac{\mathbb{E}\left[K_{\boldsymbol{N}}(x,x)^2\right]}{\mathbb{E}\left[K_{\boldsymbol{N}}(x,x)\right]^2} \simeq \exp\left(5\beta\right)\left(1 + O\left(\beta/n\right)\right), \qquad \beta = d/n,$$

*where $f \simeq g$ means $f$ is bounded above and below by universal constants times $g$.*

This result shows that in the deep and wide double scaling limit

$$n_i, d \to \infty, \qquad 0 < \lim_{n_i, d \to \infty}\sum_{i=1}^{d}\frac{1}{n_i} < \infty,$$

the NTK does *not* converge to a constant in probability. This is contrast to the wide and shallow regime where $n_i \to \infty$ and $d < \infty$ is fixed.

Our next result shows that when $\mathcal{L}$ is the square loss $K_{\boldsymbol{N}}(x,x)$ is not frozen during training. To state it, fix an input $x \in \mathbb{R}^{n_0}$ to $\boldsymbol{N}$ and define $\Delta K_{\boldsymbol{N}}(x,x)$ to be the update from one step of SGD with a batch of size $1$ containing $x$ (and learning rate $\lambda$).

**Theorem 2** (Mean of Time Derivative of NKT on Diagonal at Init). *We have that $\mathbb{E}\left[\lambda^{-1}\Delta K_{\boldsymbol{N}}(x,x)\right]$ is bounded above and below by universal constants times*

$$\left[\frac{\|x\|_2^4}{n_0^2}\sum_{\substack{i_1,i_2=1\\i_i<i_2}}^{d}\sum_{\ell=i_1}^{i_2-1}\frac{e^{-5/n_\ell-6\sum_{i=i_1}^{\ell}\frac{1}{n_i}}}{n_\ell} + \frac{\|x\|_2^2}{n_0}\sum_{\substack{i_1,i_2=1\\i_1<i_2}}^{d}e^{-5\sum_{i=1}^{i_1}\frac{1}{n_i}}\sum_{\ell=i_1}^{i_2-1}\frac{e^{-6\sum_{i=i_1+1}^{\ell-1}\frac{1}{n_i}}}{n_\ell}\right]\exp\left(5\beta\right)$$

*times a multiplicative error of size $\left(1 + O\left(\sum_{i=1}^{d}\frac{1}{n_i^2}\right)\right)$, where $\beta = \sum_{i=1}^{d}1/n_i$, as in Theorem 1. In particular, if all the hidden layer widths are equal (i.e. $n_i = n$, for $i = 1, \ldots, d-1$), we find*

$$\frac{\mathbb{E}\left[\Delta K_{\boldsymbol{N}}(x,x)\right]}{\mathbb{E}\left[K_{\boldsymbol{N}}(x,x)\right]} \simeq \frac{d\beta}{n_0}\exp\left(5\beta\right)\left(1 + O\left(\beta/n\right)\right), \qquad \beta = d/n.$$

Observe that when $d$ is fixed and $n_i = n \to \infty$, the pre-factor in front of $\exp\left(5\beta\right)$ scales like $1/n$. This is in keeping with the results from Dyer & Gur-Ari (2019) and Jacot et al. (2018). Moreover, it shows that if $d, n, n_0$ grow in any way so that $d\beta/n_0 = d^2/nn_0 \to 0$, the update $\Delta K_{\boldsymbol{N}}(x,x)$ to $K_{\boldsymbol{N}}(x,x)$ from the batch $\{x\}$ at initialization will have mean $0$. It is unclear whether this will be true also for larger batches and when the arguments of $K_{\boldsymbol{N}}$ are not equal. In contrast, if $n_i \simeq n$ and $\beta = d/n$ is bounded away from $0, \infty$, and the $n_0$ is proportional to $d$, the average update $\mathbb{E}[\Delta K_{\boldsymbol{N}}(x,x)]$ has the same order of magnitude as $\mathbb{E}[K_{\boldsymbol{N}}(x)]$.

## 2.1 ORGANIZATION FOR THE REST OF THE ARTICLE

The remainder of this article is structured as follows. First, we give an outline of the proofs of Theorems 1 and 2 in §3 and particularly in §3.1, which gives an in-depth but informal explanation of our strategy for computing moments of $K_{\boldsymbol{N}}$ and its time derivative. Next, in the Appendix Section §A, we introduce some notation about paths and edges in the computation graph of $\boldsymbol{N}$. This

notation will be used in the proofs of Theorems 1 and 2 presented in the Appendix Section §B-§D. The computations in §B explain how to handle the contribution to $K_N$ and $\Delta K_N$ coming only from the weights of the network. They are the most technical and we give them in full detail. Then, the discussion in §C and §D show how to adapt the method developed in §B to treat the contribution of biases and mixed bias-weight terms in $K_N, K_N^2$ and $\Delta K_N$. Since the arguments are simpler in these cases, we omit some details and focus only on highlighting the salient differences.

## 3 OVERVIEW OF PROOF OF THEOREMS 1 AND 2

The proofs of Theorems 1 and 2 are so similar that we will prove them at the same time. In this section and in §3.1 we present an overview of our argument. Then, we carry out the details in Appendix Sections §B-§D below. Fix an input $x \in \mathbb{R}^{n_0}$ to $N$. Recall from (5) that

$$K_N(x,x) = K_{\mathrm{w}} + K_{\mathrm{b}},$$

where we've set

$$K_{\mathrm{w}} := \sum_{\text{weights } w} \left( \frac{\partial N}{\partial w}(x) \right)^2, \qquad K_{\mathrm{b}} := \sum_{\text{biases } b} \left( \frac{\partial N}{\partial b}(x) \right)^2 \tag{6}$$

and have suppressed the dependence on $x, N$. Similarly, we have

$$-\frac{1}{2\lambda} \Delta K_N(x,x) = \Delta_{\mathrm{ww}} + 2\Delta_{\mathrm{wb}} + \Delta_{\mathrm{bb}},$$

where we have introduced

$$\Delta_{\mathrm{ww}} := \sum_{\text{weights } w,w'} \frac{\partial N}{\partial w}(x) \frac{\partial^2 N}{\partial w \partial w'}(x) \frac{\partial N}{\partial w'}(x) \left( N(x) - N_*(x) \right)$$

$$\Delta_{\mathrm{wb}} := \sum_{\text{weight } w, \text{ bias } b} \frac{\partial N}{\partial w}(x) \frac{\partial^2 N}{\partial w \partial b}(x) \frac{\partial N}{\partial b}(x) \left( N(x) - N_*(x) \right)$$

$$\Delta_{\mathrm{bb}} := \sum_{\text{biases } b,b'} \frac{\partial N}{\partial b}(x) \frac{\partial^2 N}{\partial b \partial b}(x) \frac{\partial N}{\partial b'}(x) \left( N(x) - N_*(x) \right)$$

and have used that the loss on the batch $\{x\}$ is given by $\mathcal{L}(x) = \frac{1}{2} \left( N(x) - N_*(x) \right)^2$ for some target value $N_*(x)$. To prove Theorem 1 we must estimate the following quantities:

$$\mathbb{E}[K_{\mathrm{w}}], \quad \mathbb{E}[K_{\mathrm{b}}], \quad \mathbb{E}[K_{\mathrm{w}}^2], \quad \mathbb{E}[K_{\mathrm{w}}K_{\mathrm{b}}], \quad \mathbb{E}[K_{\mathrm{b}}^2].$$

To prove Theorem 2, we must control in addition

$$\mathbb{E}[\Delta_{\mathrm{ww}}], \quad \mathbb{E}[\Delta_{\mathrm{wb}}], \quad \mathbb{E}[\Delta_{\mathrm{bb}}].$$

The most technically involved computations will turn out to be those involving only weights: namely, the terms $\mathbb{E}[K_{\mathrm{w}}], \mathbb{E}[K_{\mathrm{w}}^2], \mathbb{E}[\Delta_{\mathrm{ww}}]$. These terms are controlled by writing each as a sum over certain paths $\gamma$ that traverse the network from the input to the output layers. The corresponding results for terms involving the bias will then turn out to be very similar but with paths that start somewhere in the middle of network (corresponding to which bias term was used to differentiate the network output). The main result about the pure weight contributions to $K_N$ is the following

**Proposition 3** (Pure weight moments for $K_N, \Delta K_N$). *We have*

$$\mathbb{E}[K_{\mathrm{w}}] = \frac{d}{n_0} \|x\|_2^2.$$

*Moreover,*

$$\mathbb{E}[K_{\mathrm{w}}^2] \simeq \frac{d^2}{n_0^2} \|x\|_2^4 \exp(5\beta) \left( 1 + O\left( \sum_{i=1}^d \frac{1}{n_i^2} \right) \right), \qquad \beta := \sum_{i=1}^d \frac{1}{n_i}.$$

*Finally,*

$$\mathbb{E}[\Delta_{\mathrm{ww}}] \simeq \frac{\|x\|_2^4}{n_0^2} \left[ \sum_{\substack{i_1,i_2=1 \\ i_i < i_2}}^d \sum_{\ell=i_1}^{i_2-1} \frac{1}{n_\ell} e^{-5/n_\ell - 6 \sum_{i=i_1}^{\ell-1} \frac{1}{n_i}} \right] \exp(5\beta) \left( 1 + O\left( \sum_{i=1}^d \frac{1}{n_i^2} \right) \right).$$

We prove Proposition 3 in §B below. The proof already contains all the ideas necessary to treat the remaining moments. In §C and §D we explain how to modify the proof of Proposition 3 to prove the following two Propositions:

**Proposition 4** (Pure bias moments for $K_N, \Delta K_N$). *We have*

$$\mathbb{E}[K_{\mathrm{b}}] \;=\; \frac{d}{2}.$$

*Moreover,*

$$\mathbb{E}[K_{\mathrm{b}}^2] \;\simeq\; \left[\sum_{\substack{i,j=1 \\ i \le j}}^{d} e^{-5 \sum_{\ell=1}^{j} \frac{1}{n_i}}\right] \exp\left(5 \sum_{i=1}^{d} \frac{1}{n_i}\right)\left(1 + O\left(\sum_{i=1}^{d} \frac{1}{n_i^2}\right)\right).$$

*Finally, with probability* 1, *we have* $\Delta_{\mathrm{bb}} = 0$.

**Proposition 5** (Mixed bias-weight moments for $K_N, \Delta K_N$). *We have*

$$\mathbb{E}[K_{\mathrm{b}} K_{\mathrm{w}}] \;\simeq\; \frac{d \left\|x\right\|_2^2}{n_0}\left[\sum_{j=1}^{d} e^{-5 \sum_{i=1}^{j} \frac{1}{n_i}}\right] \exp\left(5 \sum_{i=1}^{d} \frac{1}{n_i}\right)\left(1 + O\left(\sum_{i=1}^{d} \frac{1}{n_i^2}\right)\right).$$

*Further,* $\mathbb{E}[\Delta_{\mathrm{wb}}]$ *is bounded above and below by universal constants times*

$$\frac{\left\|x\right\|_2^2}{n_0} \exp\left(5 \sum_{i=1}^{d} \frac{1}{n_i}\right)\left[\sum_{\substack{i,j=1 \\ j<i}}^{d} e^{-5 \sum_{\alpha=1}^{j} \frac{1}{n_\alpha}} \sum_{\ell=j}^{i-1} \frac{1}{n_\ell} e^{-6 \sum_{\alpha=j+1}^{\ell-1} \frac{1}{n_\alpha}}\right]\left(1 + O\left(\sum_{i=1}^{d} \frac{1}{n_i^2}\right)\right).$$

The statements in Theorems 1 and 2 that hold for general $n_i$ now follow directly from Propositions 3-5. The asymptotics when $n_i \simeq n$ follow from some routine algebra.

## 3.1 IDEA OF PROOF OF PROPOSITIONS 3-5

Before turning to the details of the proof of Propositions 3-5 below, we give an intuitive explanation of the key steps in our sum-over-path analysis of the moments of $K_{\mathrm{w}}, K_{\mathrm{b}}, \Delta_{\mathrm{ww}}, \Delta_{\mathrm{wb}}, \Delta_{\mathrm{bb}}$. Since the proofs of all three Propositions follow a similar structure and Proposition 3 is the most complicated, we will focus on explaining how to obtain the first 2 moments of $K_{\mathrm{w}}$. Since the biases are initialized to zero and $K_{\mathrm{w}}$ involves only derivatives with respect to the weights, for the purposes of analyzing $K_{\mathrm{w}}$ the biases play no role. Without the biases, the output of the neural network, $\boldsymbol{N}(x)$ can be express as a weighted sum over paths in the computational graph of the network:

$$\boldsymbol{N}(x) \;=\; \sum_{a=1}^{n_0} x_a \sum_{\gamma \in \Gamma_a^1} \mathrm{wt}(\gamma),$$

where $\Gamma_a^1$ is the collection of paths in $\boldsymbol{N}$ starting at neuron $a$ and the weight of a path $\mathrm{wt}(\gamma)$ is defined in (13) in the Appendix and includes both the product of the weights along $\gamma$ and the condition that every neuron in $\gamma$ is open at $x$. The path $\gamma$ begins at some neuron in the input layer of $\boldsymbol{N}$ and passes through a neuron in every subsequent layer until ending up at the unique neuron in the output layer (see (10)). Being a product over edge weights in a given path, the derivative of $\mathrm{wt}(\gamma)$ with respect to a weight $W_e$ on an edge $e$ of the computational graph of $\boldsymbol{N}$ is:

$$\frac{\partial \mathrm{wt}(\gamma)}{\partial W_e} = \frac{\mathrm{wt}(\gamma)}{W_e} \mathbf{1}_{\{e \in \gamma\}}. \tag{7}$$

There is a subtle point here that $\mathrm{wt}(\gamma)$ also involves indicator functions of the events that neurons along $\gamma$ are open at $x$. However, with probability 1, the derivative with respect to $W_e$ of these indicator functions is identically 0 at $x$. The details are in Lemma 11.

Because $K_{\mathrm{w}}$ is a sum of derivatives squared (see (6)), ignoring the dependence on the network input $x$, the kernel $K_{\mathrm{w}}$ roughly takes the form

$$K_{\mathrm{w}} \sim \sum_{\gamma_1, \gamma_2} \sum_{e \in \gamma_1 \cap \gamma_2} \frac{\prod_{k=1}^{2} \mathrm{wt}(\gamma_k)}{W_e^2},$$

where the sum is over collections $(\gamma_1, \gamma_2)$ of two paths in the computation graph of $N$ and edges $e$ in the computational graph of $N$ that lie on both (see Lemma 6 for the precise statement). When computing the mean, $\mathbb{E}[K_{\mathrm{w}}]$, by the mean zero assumption of the weights $W_e$ (see (4)), the only contribution is when every edge in the computational graph of $N$ is traversed by an even number of paths. Since there are exactly two paths, the only contribution is when the two paths are identical, dramatically simplifying the problem. This gives rise to the simple formula for $\mathbb{E}[K_{\mathrm{w}}]$ (see (23)). The expression

$$K_{\mathrm{w}}^2 \sim \sum_{\gamma_1, \gamma_2, \gamma_3, \gamma_4} \sum_{\substack{e_1 \in \gamma_1 \cap \gamma_2 \\ e_2 \in \gamma_3 \cap \gamma_4}} \frac{\prod_{k=1}^{4} \mathrm{wt}(\gamma_k)}{W_{e_1}^2 W_{e_2}^2},$$

for $K_{\mathrm{w}}^2$ is more complex. It involves sums over four paths in the computational graph of $N$ as in the second statement of Lemma 6. Again recalling that the moments of the weights have mean $0$, the only collections of paths that contribute to $\mathbb{E}[K_{\mathrm{w}}^2]$ are those in which every edge in the computational graph of $N$ is covered an even number of times:

$$\mathbb{E}[K_{\mathrm{w}}^2] \sim \sum_{\substack{\gamma_1, \gamma_2, \gamma_3, \gamma_4 \\ \mathrm{even}}} \sum_{\substack{e_1 \in \gamma_1 \cap \gamma_2 \\ e_2 \in \gamma_3 \cap \gamma_4}} \mathbb{E}\left[\frac{\prod_{k=1}^{4} \mathrm{wt}(\gamma_k)}{W_{e_1}^2 W_{e_2}^2}\right] \tag{8}$$

However, there are now several ways the four paths can interact to give such a configuration. It is the combinatorics of these interactions, together with the stipulation that the marked edges $e_1, e_2$ belong to particular pairs of paths, which complicates the analysis of $\mathbb{E}[K_{\mathrm{w}}^2]$. We estimate this expectation in several steps:

1. Obtain an exact formula for the expectation in (8):

$$\mathbb{E}\left[\frac{\prod_{k=1}^{4} \mathrm{wt}(\gamma_k)}{W_{e_1}^2 W_{e_2}^2}\right] = F(\Gamma, e_1, e_2),$$

   where $F(\Gamma, e_1, e_2)$ is the product over the layers $\ell = 1, \ldots, d$ in $N$ of the "cost" of the interactions of $\gamma_1, \ldots, \gamma_4$ between layers $\ell - 1$ and $\ell$. The precise formula is in Lemma 7.

2. Observe the dependence of $F(\Gamma, e_1, e_2)$ on $e_1, e_2$ is only up to a multiplicative constant:

$$F(\Gamma, e_1, e_2) \simeq F_*(\Gamma).$$

   The precise relation is (24). This shows that, up to universal constants,

$$\mathbb{E}[K_{\mathrm{w}}^2] \simeq \sum_{\substack{\gamma_1, \gamma_2, \gamma_3, \gamma_4 \\ \mathrm{even}}} F_*(\Gamma) \# \left\{\ell_1, \ell_2 \in [d] \ \middle| \ \begin{array}{l} \gamma_1, \gamma_2 \text{ togethe at layer } \ell_1 \\ \gamma_3, \gamma_4 \text{ togethe at layer } \ell_2 \end{array}\right\}.$$

   This is captured precisely by the terms $I_j, II_j$ defined in (27),(28).

3. Notice that $F_*(\Gamma)$ depends only on the un-ordered multiset of edges $E = E^\Gamma \in \Sigma_{even}^4$ determined by $\Gamma$ (see (17) for a precise definition). We therefore change variables in the sum from the previous step to find

$$\mathbb{E}[K_{\mathrm{w}}^2] \simeq \sum_{E \in \Sigma_{even}^4} F_*(E) \mathrm{Jacobian}(E, e_1, e_2),$$

   where $\mathrm{Jacobian}(E, e_1, e_2)$ counts how many collections of four paths $\Gamma \in \Gamma_{even}^4$ that have the same $E^\Gamma$ also have paths $\gamma_1, \gamma_2$ pass through $e_1$ and paths $\gamma_3, \gamma_4$ pass through $e_2$. Lemma 8 gives a precise expression for this Jacobian. It turns out, as explained just below Lemma 8, that

$$\mathrm{Jacobian}(E, e_1, e_2) \simeq 6^{\#\mathrm{loops}(E)},$$

   where a loop in $E$ occurs when the four paths interact. More precisely, a loop occurs whenever all four paths pass through the same neuron in some layer (see Figures 1 and 2).

4. Change variables from unordered multisets of edges $E \in \Sigma_{even}^4$ in which every edge is covered an even number of times to pairs of paths $V \in \Gamma^2$. The Jacobian turns out to be $2^{-\#\mathrm{loops}(E)}$ (Lemma 9), giving

$$\mathbb{E}[K_{\mathrm{w}}^2] \simeq \sum_{V \in \Gamma^2} F_*(V) 3^{\#\mathrm{loops}(V)}.$$

5. Just like $F_*(V)$, the term $3^{\#\mathrm{loops}(V)}$ is again a product over layers $\ell$ in the computational graph of $N$ of the "cost" of interactions between our four paths. Aggregating these two terms into a single functional $\widehat{F}_*(E)$ and factoring out the $1/n_\ell$ terms in $F_*(V)$ we find that:

$$\mathbb{E}[K_{\mathrm{w}}^2] \simeq \frac{1}{n_0^2} \mathcal{E}\left[\widehat{F}_*(V)\right],$$

where the $1/n_\ell$ terms cause the sum to become *an average* over collections $V$ of two independent paths in the computational graph of $N$, with each path sampling neurons uniformly at random in every layer. The precise result, including the dependence on the input $x$, is in (42).

6. Finally, we use Proposition 10 to obtain for this expectation estimates above and below that match up multiplicative constants.

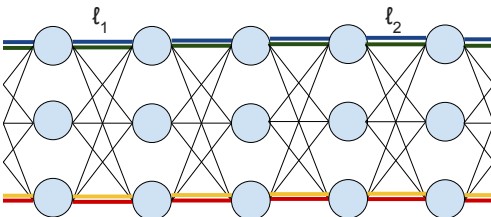

Figure 1: Cartoon of the four paths $\gamma_1, \gamma_2, \gamma_3, \gamma_4$ between layers $\ell_1$ and $\ell_2$ in the case where there is no interaction. Paths stay with there original partners $\gamma_1$ with $\gamma_2$ and $\gamma_3$ with $\gamma_4$ at all intermediate layers.

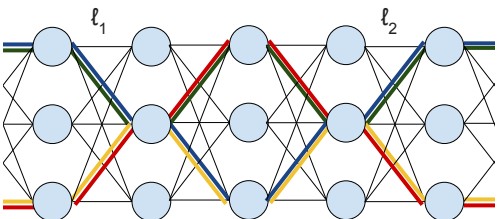

Figure 2: Cartoon of the four paths $\gamma_1, \gamma_2, \gamma_3, \gamma_4$ between layers $\ell_1$ and $\ell_2$ in the case where there is exactly one "loop" interaction between the marked layers. Paths swap away from their original partners exactly once at some intermediate layer after $\ell_1$, and then swap back to their original partners before $\ell_2$.

## 3.2 CONCLUSION

Taken together Theorems 1 and 2 show that in fully connected ReLU nets that are both deep and wide the neural tangent kernel $K_N$ is genuinely stochastic and enjoys a non-trivial evolution during training. This suggests that in the overparameterized limit $n, d \to \infty$ with $d/n \in (0, \infty)$, the kernel $K_N$ may learn data-dependent features. Moreover, our results show that the fluctuations of both $K_N$ and its time derivative are exponential in the inverse temperature $\beta = d/n$.

It would be interesting to obtain an exact description of its statistics at initialization and to describe the law of its trajectory during training. Assuming this trajectory turns out to be data-dependent,

our results suggest that the double descent curve Belkin et al. (2018; 2019); Spigler et al. (2018) that trades off complexity vs. generalization error may display significantly different behaviors depending on the mode of network overparameterization.

However, it is also important to point out that the results in Hanin (2018); Hanin & Nica (2018); Hanin & Rolnick (2018) show that, at least for fully connected ReLU nets, gradient-based training is not numerically stable unless $d/n$ is relatively small (but not necessarily zero). Thus, we conjecture that there may exist a "weak feature learning" NTK regime in which network depth and width are both large but $0 < d/n \ll 1$. In such a regime, the network will be stable enough to train but flexible enough to learn data-dependent features. In the language of Chizat & Bach (2018b) one might say this regime displays weak lazy training in which the model can still be described by a stochastic positive definite kernel whose fluctuations can interact with data.

Finally, it is an interesting question to what extent our results hold for non-linearities other than ReLU and for network architectures other than fully connected (e.g. convolutional and residual). Even in fully connected networks, the input/outpu Jacobian already displays different spectral statistics depending on the non-linearity (Pennington et al., 2017). Moreover, typical ConvNets, are significantly wider than they are deep, and we leave it to future work to adapt the techniques from the present article to these more general settings (the neural tangent kernel for finite depth convolutional networks is studied in part in (Arora et al., 2019)).

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

## A    NOTATION

In this section, we introduce some notation, adapted in large part from Hanin & Nica (2018), that will be used in the proofs of Theorems 1 and 2. For $n \in \mathbb{N}$, we will write

$$[n] := \{1, \ldots, n\}.$$

It will also be convenient to denote

$$[n]_{even}^k := \{a \in [n]^k \mid \text{every entry in } a \text{ appears an even number of times}\}.$$

Given a ReLU network $\boldsymbol{N}$ with input dimension $n_0$, hidden layer widths $n_1, \ldots, n_{d-1}$, and output dimension $n_d = 1$, its *computational graph* is a directed multipartite graph whose vertex set is the disjoint union $[n_0] \coprod \cdots \coprod [n_d]$ and in which edges are all possible ways of connecting vertices from $[n_{i-1}]$ with vertices from $[n_i]$ for $i = 1, \ldots, d$. The vertices are the *neurons* in $\boldsymbol{N}$, and we will write for $\ell \in \{0, \ldots, d\}$ and $\alpha \in [n_\ell]$

$$z(\ell, \alpha) := \text{ neuron number } \alpha \text{ in layer } \ell. \tag{9}$$

**Definition 1** (Path in the computational graph of $\boldsymbol{N}$). *Given $0 \le \ell_1 < \ell_2 \le d$ and $\alpha_1 \in [n_{\ell_1}]$, $\alpha_2 \in [n_{\ell_2}]$, a path $\gamma$ in the computational graph of $\boldsymbol{N}$ from neuron $z(\ell_1, \alpha_1)$ to neuron $z(\ell_2, \alpha_2)$ is a collection of neurons in layers $\ell_1, \ldots, \ell_2$:*

$$\gamma = (\gamma(\ell_1), \ldots, \gamma(\ell_2)), \qquad \gamma(j) \in [n_j], \quad \gamma(\ell_1) = \alpha_1, \gamma(\ell_2) = \alpha_2. \tag{10}$$

Further, we will write
$$Z^k = \{(z_1, \ldots, z_k) \mid z_j \text{ are neurons in } \boldsymbol{N}\}.$$
Given a collection of neurons
$$Z = (z(\ell_1, \alpha_1), \ldots, z(\ell_k, \alpha_k)) \in Z^k$$
we denote by
$$\Gamma_Z^k := \left\{ (\gamma_1, \ldots, \gamma_k) \mid \begin{smallmatrix} \gamma_j \text{ is a path starting at neuron } z(\ell_j, \alpha_j) \\ \text{ending at the output neuron } z(d, 1) \end{smallmatrix} \right\}$$
Note that with this notation, we have $\gamma_i \in \Gamma^1_{z(\ell_i, \alpha_i)}$ for each $i = 1, \ldots, k$. For $\Gamma \in \Gamma_Z^k$, we also set
$$\Gamma(\ell) = \{\alpha \in [n_\ell] \mid \exists j \in [k] \text{ s.t. } \gamma_j(\ell) = k\}.$$
Correspondingly, we will write
$$|\Gamma(\ell)| := \# \text{ distinct elements in } \Gamma(\ell). \tag{11}$$

If each edge $e$ in the computational graph of $\boldsymbol{N}$ is assigned a weight $\widehat{W}_e$, then associated to a path $\gamma$ is a collection of weights:
$$\widehat{W}_\gamma^{(i)} := \widehat{W}_{(\gamma(i-1), \gamma(i))}. \tag{12}$$

**Definition 2** (Weight of a path in the computational graph of $\boldsymbol{N}$). *Fix $0 \leq \ell \leq d$, and let $\gamma$ be a path in the computation graph of $\boldsymbol{N}$ starting at layer $\ell$ and ending at the output. The weight of a this path at a given input $x$ to $\boldsymbol{N}$ is*

$$\mathrm{wt}(\gamma) := \widehat{W}_\gamma^{(d)} \prod_{j=\ell+1}^{d-1} \widehat{W}_\gamma^{(j)} \mathbf{1}_{\{\gamma \text{ open at } x\}}, \tag{13}$$

*where*

$$\mathbf{1}_{\{\gamma \text{ open at } x\}} = \prod_{i=\ell}^{d} \xi_\gamma^{(i)}(x), \qquad \xi_\gamma^{(\ell)}(x) := \mathbf{1}_{\{y_\gamma^{(\ell)} > 0\}},$$

*is the event that all neurons along $\gamma$ are open for the input $x$. Here $y^{(\ell)}$ is as in (2).*

Next, for an edge $e \in [n_{i-1}] \times [n_i]$ in the computational graph of $\boldsymbol{N}$ we will write
$$\ell(e) = i \tag{14}$$
for the layer of $e$. In the course of proving Theorems 1 and 2, it will be useful to associate to every $\Gamma \in \Gamma^k(\vec{n})$ an *unordered multi-set* of edges $E^\Gamma$.

**Definition 3** (Unordered multisets of edges and their endpoints). *For $n, n', \ell \in \mathbb{N}$ set*
$$\Sigma^k(n, n') = \{(\alpha_1, \beta_1), \ldots, (\alpha_k, \beta_k) \mid (\alpha_j, \beta_j) \in [n] \times [n']\}$$
*to be the unordered multiset of edges in the complete directed bi-paritite graph $K_{n,n'}$ oriented from $[n]$ to $[n']$. For every $E \in \Sigma^k(n, n')$ define its left and right endpoints to be*
$$L(E) := \{\alpha \in [n] \mid \exists j = 1, \ldots, k \text{ s.t. } \alpha = \alpha_j\} \tag{15}$$
$$R(E) := \{\beta \in [n'] \mid \exists j = 1, \ldots, k \text{ s.t. } \beta = \beta_j\}, \tag{16}$$
*where $L(E), R(E)$ are unordered multi-sets.*

Using this notation, for any collection $Z = (z(\ell_1, \alpha_1), \ldots, z(\ell_k, \alpha_k))$ of neurons and $\Gamma = (\gamma_1, \ldots, \gamma_k) \in \Gamma_Z^k$, define for each $\ell \in [d]$ the associated unordered multiset
$$E^\Gamma(\ell) := \{(\alpha, \beta) \in [n_{\ell-1}, n_\ell] \mid \exists j = 1, \ldots, k \text{ s.t. } \gamma_j(\ell-1) = \alpha, \gamma_j(\ell) = \beta\}$$
of edges between layers $\ell - 1$ and $\ell$ that are present in $\Gamma$. Similarly, we will write
$$\Sigma_Z^k := \{(E(0), \ldots, E(d)) \in \Sigma^k(n_0, n_1) \times \cdots \times \Sigma^k(n_{d-1}, n_d) \mid \exists \Gamma \in \Gamma_Z^k \text{ s.t. } E(\ell) = E^\Gamma(\ell), \ \ell \in [d]\} \tag{17}$$
for the set of all possible edge multisets realized by paths in $\Gamma_Z^k$. On a number of occasions, we will also write
$$\Sigma_{Z,even}^k := \{E \in \Sigma_Z^k \mid \text{ every edge in } E \text{ appears an even number of times}\}$$

and correspondingly

$$\Gamma_{Z,even}^k := \{\Gamma \in \Gamma_Z^k \mid E^\Gamma \in \Sigma_{Z,even}^k\}.$$

We will moreover say that for a path $\gamma$ an edge $e = (\alpha, \beta) \in [n_{i-1}] \times [n_i]$ in the computational graph of $\boldsymbol{N}$ belongs to $\gamma$ (written $e \in \gamma$) if

$$\gamma(i-1) = \alpha, \qquad \gamma(i) = \beta. \tag{18}$$

Finally, for an edge $e = (\alpha, \beta) \in [n_{i-1}] \times [n_i]$ in the computational graph of $\boldsymbol{N}$, we set

$$W_e = W_{\alpha,\beta}^{(i)}, \qquad \widehat{W}_e = \widehat{W}_{\alpha,\beta}^{(i)}$$

for the normalized and unnormalized weights on the edge corresponding to $e$ (see (3)).

## B   PROOF OF PROPOSITION 3

We begin with the well-known formula for the output of a ReLU net $\boldsymbol{N}$ with biases set to $0$ and a linear final layer with one neuron:

$$\boldsymbol{N}(x) = \sum_{a=1}^{n_0} x_a \sum_{\gamma \in \Gamma_a^1} \text{wt}(\gamma). \tag{19}$$

The weight of a path $\text{wt}(\gamma)$ was defined in (13) and includes both the product of the weights along $\gamma$ and the condition that every neuron in $\gamma$ is open at $x$. As explained in §A, the inner sum in (19) is over paths $\gamma$ in the computational graph of $\boldsymbol{N}$ that start at neuron $a$ in the input layer and end at the output neuron and the random variables $\widehat{W}_\gamma^{(i)}$ are the normalized weights on the edge of $\gamma$ between layer $i-1$ and layer $i$ (see (12)). Differentiating this formula gives sum-over-path expressions for the derivatives of $\boldsymbol{N}$ with respect to both $x$ and its trainable parameters. For the NTK and its first SGD update, the result is the following:

**Lemma 6** (weight contribution to $K_{\boldsymbol{N}}$ and $\Delta K_{\boldsymbol{N}}$ as a sum-over-paths). *With probability* $1$,

$$K_{\text{w}} = \sum_{\substack{a \in [n_0]^2}} \prod_{k=1}^2 x_{a_k} \sum_{\substack{\Gamma \in \Gamma_a^2 \\ \Gamma = (\gamma_1, \gamma_2)}} \sum_{e \in \gamma_1 \cap \gamma_2} \frac{\prod_{k=1}^2 \text{wt}(\gamma_k)}{W_e^2},$$

*where the sum is over collections $\Gamma$ of two paths in the computation graph of $\boldsymbol{N}$ and edges $e$ that lie on both paths. Similarly, almost surely,*

$$K_{\text{w}}^2 = \sum_{\substack{a \in [n_0]^4}} \prod_{k=1}^4 x_{a_k} \sum_{\substack{\Gamma \in \Gamma_a^4(\vec{n}) \\ \Gamma = (\gamma_1, \ldots, \gamma_4)}} \sum_{\substack{e_1 \in \gamma_1 \cap \gamma_2 \\ e_2 \in \gamma_3, \gamma_4}} \frac{\prod_{k=1}^4 \text{wt}(\gamma_k)}{W_{e_1}^2 W_{e_2}^2},$$

*and*

$$\Delta_{\text{ww}} = \sum_{\substack{a \in [n_0]^4}} \prod_{k=1}^4 x_{a_k} \sum_{\substack{\Gamma \in \Gamma_a^4(\vec{n}) \\ \Gamma = (\gamma_1, \ldots, \gamma_4)}} \sum_{\substack{e_1 \in \gamma_1 \cap \gamma_2 \\ e_2 \in \gamma_2, \gamma_3 \\ e_1 \neq e_2}} \frac{\prod_{k=1}^4 \text{wt}(\gamma_k)}{W_{e_1}^2 W_{e_2}^2}$$

*plus a term that has mean* $0$.

The notation $[n_0]^k$, $\Gamma_a^k$, $e \in \gamma$, etc is defined in §A. We prove Lemma 6 in §B.1 below. Let us emphasize that the expressions for $K_{\text{w}}^2$ and $\Delta_{\text{ww}}$ are almost identical. The main difference is that in the expression for $\Delta_{\text{ww}}$, the second path $\gamma_2$ must contain both $e_1$ and $e_2$ while $\gamma_4$ has no restrictions. Hence, while for $K_{\text{w}}^2$ the contribution from a collection of four paths $\Gamma = (\gamma_1, \gamma_2, \gamma_3, \gamma_4)$ is the same as from the collection $\Gamma' = (\gamma_2, \gamma_1, \gamma_4, \gamma_3)$, for $\Delta_{\text{ww}}$ the contributions are different. This seemingly small discrepancy, as we shall see, causes the normalized expectation $\mathbb{E}[\Delta_{\text{ww}}]/\mathbb{E}[K_{\text{w}}]$ to converge to zero when $d < \infty$ is fixed and $n_i \to \infty$ (see the $1/n_\ell$ factors in the statement of Theorem 2). In contrast, in the same regime, the normalized second moment $\mathbb{E}[K_{\text{w}}^2]/\mathbb{E}[K_{\text{w}}]^2$ remains bounded away from zero as in the statement of Theorem 1. Both statements are consistent with prior results in the literature Dyer & Gur-Ari (2019); Jacot et al. (2018). Taking expectations in Lemma 6 yields the following result.

**Lemma 7** (Expectation of $K_{\mathrm{w}}, K_{\mathrm{w}}^2, \Delta_{\mathrm{ww}}$ as sums over 2, 4 paths)**.** *We have,*

$$\mathbb{E}[K_{\mathrm{w}}] \;=\; \sum_{a \in [n_0]^2_{even}} \prod_{k=1}^{2} x_{a_k} \sum_{\substack{\Gamma_a \in \Gamma^2_{even} \\ \Gamma = (\gamma_1, \gamma_2)}} \sum_{e \in \gamma_1 \cap \gamma_2} H(\Gamma, e) \tag{20}$$

*where*

$$H(\Gamma, e) \;=\; \mathbf{1}_{\{\gamma_1 = \gamma_2\}} \prod_{i=1}^{d} \frac{1}{n_{i-1}}$$

*Similarly,*

$$\mathbb{E}[K_{\mathrm{w}}^2] \;=\; \sum_{a \in [n_0]^4_{even}} \prod_{k=1}^{4} x_{a_k} \sum_{\substack{\Gamma \in \Gamma^4_{a,even} \\ \Gamma = (\gamma_1, \ldots, \gamma_4)}} \sum_{\substack{e_1 \in \gamma_1 \cap \gamma_2 \\ e_2 \in \gamma_3 \cap \gamma_4}} F(\Gamma, e_1, e_2), \tag{21}$$

*where*

$$F(\Gamma, e_1, e_2) \;=\; \frac{1}{2} \prod_{i=1}^{d} \frac{2^{2-|\Gamma(i)|}}{n_{i-1}^2} \prod_{i \neq \ell(e_1), \ell(e_2)} \mu_4^{\mathbf{1}_{\{|\Gamma(i-1)| = |\Gamma(i)| = 1\}}}.$$

*Finally,*

$$\mathbb{E}[\Delta_{\mathrm{ww}}] \;=\; \sum_{a \in [n_0]^4_{even}} \prod_{k=1}^{4} x_{a_k} \sum_{\substack{\Gamma \in \Gamma^4_{a,even} \\ \Gamma = (\gamma_1, \ldots, \gamma_4)}} \sum_{\substack{e_1 \in \gamma_1 \cap \gamma_2 \\ e_2 \in \gamma_2, \gamma_3 \\ e_1 \neq e_2}} F(\Gamma, e_1, e_2). \tag{22}$$

Lemma 7 is proved in §B.2. The expression (20) is simple to evaluate due to the delta function in $H(\Gamma, e)$. We obtain:

$$\mathbb{E}[K_{\mathrm{w}}] \;=\; \sum_{a=1}^{n_0} x_a^2 \sum_{\gamma \in \Gamma^1(\vec{n})} \sum_{e \in \gamma} \prod_{i=1}^{d} \frac{1}{n_{i-1}} \;=\; d \prod_{i=1}^{d} \frac{1}{n_{i-1}} \prod_{i=1}^{d} n_i \, \|x\|_2^2 \;=\; \frac{d}{n_0} \, \|x\|_2^2, \tag{23}$$

where in the second-to-last equality we used that the number of paths in the comutational graph of $N$ from a given neuron in the input to the output neuron equals $\prod_{i=1,\ldots,d} n_i$ and in the last equality we used that $n_d = 1$. This proves the first equality in Theorem 1.

It therefore remains to evaluate (21) and (22). Since they are so similar, we will continue to discuss them in parallel. To start, notice that the expression $F(\Gamma, e_1, e_2)$ appearing in (21) and (22) satisfies

$$\frac{1}{2\mu_4^2} F_*(\Gamma) \;\leq\; F(\Gamma, e_1, e_2) \;\leq\; \frac{1}{2} F_*(\Gamma),$$

where

$$F_*(\Gamma) \;:=\; \prod_{i=1}^{d} \frac{2^{2-|\Gamma(i)|}}{n_{i-1}^2} \mu_4^{\mathbf{1}_{\{|\Gamma(i-1)| = |\Gamma(i)| = 1\}}}. \tag{24}$$

For the remainder of the proof we will write

$$f \;\simeq\; g \quad \Longleftrightarrow \quad \exists \text{ constants } C, c > 0 \text{ depending only on } \mu \text{ s.t.} \quad cg \;\leq\; f \;\leq\; Cg.$$

Thus, in particular,

$$F(\Gamma, e_1, e_2) \;\simeq\; F_*(\Gamma).$$

The advantage of $F_*(\Gamma)$ is that it does not depend on $e_1, e_2$. Observe that for every $a = (\alpha_1, \alpha_2, \alpha_3, \alpha_4) \in [n_0]^4_{even}$, we have that either $\alpha_1 = \alpha_2$, $\alpha_1 = \alpha_3$, or $\alpha_1 = \alpha_4$. Thus, by symmetry, the sum over $\Gamma^4_{even}(\vec{n})$ in (21) and (22) takes only four distinct values, represented by the following possibilities:

$$a_j \in [n_0]^4_{even} \;:=\; \begin{cases} (1,1,1,1), & j = 1 \\ (1,2,1,2), & j = 2 \\ (1,1,2,2), & j = 3 \\ (1,2,2,1), & j = 4 \end{cases},$$

keeping track of which paths $\gamma_1, \ldots, \gamma_4$ begin at the same neuron in the input layer to $N$. Hence, since

$$\sum_{\substack{a=(a_1,\ldots,a_4)\in[n_0]^4_{even}\\ a_1=a_2,\, a_3=a_4,\, a_1\neq a_3}} \prod_{k=1}^{4} x_{a_k} \;=\; \|x\|_2^4 - \|x\|_4^4$$

we find

$$\mathbb{E}[K_{\mathrm{w}}^2] \;\simeq\; \|x\|_4^4\, I_1 + (\|x\|_2^4 - \|x\|_4^4)(I_2 + I_3 + I_4), \tag{25}$$

and similarly,

$$\mathbb{E}[\Delta_{\mathrm{ww}}] \;\simeq\; \|x\|_4^4\, II_1 + (\|x\|_2^4 - \|x\|_4^4)(II_2 + II_3 + II_4), \tag{26}$$

where

$$I_j \;=\; \sum_{\substack{\Gamma\in\Gamma^4_{a_j,even}(\vec{n})\\ \Gamma=(\gamma_1,\ldots,\gamma_4)}} F_*(\Gamma)\# \left\{\text{edges } e_1, e_2 \mid e_1 \in \gamma_1 \cap \gamma_2,\, e_2 \in \gamma_3 \cap \gamma_4\right\} \tag{27}$$

$$II_j \;=\; \sum_{\substack{\Gamma\in\Gamma^4_{a_j,even}(\vec{n})\\ \Gamma=(\gamma_1,\ldots,\gamma_4)}} F_*(\Gamma)\# \left\{\text{edges } e_1, e_2 \mid e_1 \in \gamma_1 \cap \gamma_2,\, e_2 \in \gamma_2, \gamma_3,\, e_1 \neq e_2\right\}. \tag{28}$$

To evaluate $I_j$, $II_j$ let us write

$$T_i^{\alpha,\beta}(\Gamma) \;:=\; \mathbf{1}_{\left\{\substack{\gamma_\alpha(i-1)=\gamma_\beta(i-1)\\ \gamma_\alpha(i)=\gamma_\beta(i)}\right\}}, \qquad \Gamma=(\gamma_1,\ldots,\gamma_4), \qquad \alpha,\beta=1,\ldots,4 \tag{29}$$

for the indicator function of the event that paths $\gamma_\alpha, \gamma_\beta$ pass through the same edge between layers $i-1, i$ in the computational graph of $N$. Observe that

$$\# \left\{\text{edges } e_1, e_2 \mid e_1 \in \gamma_1 \cap \gamma_2,\, e_2 \in \gamma_3 \cap \gamma_4\right\} \;=\; \sum_{i_1,i_2=1}^{d} T_{i_1}^{1,2} T_{i_2}^{3,4}$$

and

$$\# \left\{\text{edges } e_1, e_2 \mid e_1 \in \gamma_1 \cap \gamma_2,\, e_2 \in \gamma_2, \gamma_3,\, e_1 \neq e_2\right\} \;=\; \sum_{\substack{i_1,i_2=1\\ i_1\neq i_2}}^{d} T_{i_1}^{1,2} T_{i_2}^{2,3}.$$

Thus, we have

$$I_j \;=\; \sum_{i_1,i_2=1}^{d} I_{j,i_1,i_2}, \qquad II_j \;=\; \sum_{\substack{i_1,i_2=1\\ i_1\neq i_2}}^{d} II_{j,i_1,i_2},$$

where

$$I_{j,i_1,i_2} \;=\; \sum_{\substack{\Gamma\in\Gamma^4_{a_j,even}\\ \Gamma=(\gamma_1,\ldots,\gamma_4)}} F_*(\Gamma) T_{i_1}^{1,2} T_{i_2}^{3,4}, \qquad II_{j,i_1,i_2} \;=\; \sum_{\substack{\Gamma\in\Gamma^4_{a_j,even}\\ \Gamma=(\gamma_1,\ldots,\gamma_4)}} F_*(\Gamma) T_{i_1}^{1,2} T_{i_2}^{2,3}.$$

To simplify $I_{j,i_1,i_2}$ and $II_{j,i_1,i_2}$ observe that $F_*(\Gamma)$ depends only on $\Gamma$ only via the unordered edge multi-set (i.e. only which edges are covered matters; not their labelling)

$$E^\Gamma = \left(E^\Gamma(1),\ldots,E^\Gamma(d)\right) \in \Sigma^4_{even}$$

defined in Definition 3. Hence, we find that for $j=1,2,3,4$, $i_1,i_2=1,\ldots,d$,

$$I_{j,i_1,i_2} \;=\; \sum_{E\in\Sigma^4_{a_j,even}(\vec{n})} F_*(E)\# \left\{\Gamma\in\Gamma^4_{a_j,even}(\vec{n}) \mid {\substack{E^\Gamma=E,\; \Gamma(0)=a_j,\, t=1,2\\ \gamma_1(i_t-1)=\gamma_2(i_t-1),\, \gamma_1(i_t)=\gamma_2(i_t)}}\right\} \tag{30}$$

$$II_{j,i_1,i_2} \;=\; \sum_{E\in\Sigma^4_{a_j,even}(\vec{n})} F_*(E)\# \left\{\Gamma\in\Gamma^4_{a_j,even}(\vec{n}) \;\middle|\; {\substack{E^\Gamma=E,\; \Gamma(0)=a_j\\ \gamma_1(i_1-1)=\gamma_2(i_1-1),\, \gamma_1(i_1)=\gamma_2(i_1)\\ \gamma_2(i_2-1)=\gamma_3(i_2-1),\, \gamma_2(i_2)=\gamma_3(i_2)}}\right\} \tag{31}$$

The counts in $I_{j,*,i_1,i_2}$ and $II_{j,*,i_1,i_2}$ have a convenient representation in terms of

$$C(E, i_1, i_2) := \mathbf{1}_{\{\exists\, \ell = \min(i_1,i_2),\ldots,\max(i_1,i_2-1)\ \text{s.t.}\ |R(E(\ell))|=1\}} \tag{32}$$

$$\widehat{C}(E, i_1, i_2) := \mathbf{1}_{\{\exists\, \ell = 0,\ldots,\min(i_1,i_2)-1\ \text{s.t.}\ |R(E(\ell))|=1\}}. \tag{33}$$

Informally, the event $\widehat{C}(E, i_1, i_2)$ indicates the presence of a "collision" of the four paths in $\Gamma$ before the earlier of the layers $i_1, i_2$, while $C(E, i_1, i_2)$ gives a "collision" between layers $i_1, i_2$; see Section 3.1 for the intuition behind calling these collisions. We also write

$$A(E, i_1, i_2) := \mathbf{1}_{\left\{\begin{smallmatrix}|L(E(i_1))|=|R(E(i_1))|=1\\|L(E(i_2))|=|R(E(i_2))|=1\end{smallmatrix}\right\}} + \frac{1}{6}\mathbf{1}_{\left\{\begin{smallmatrix}|L(E(i_1))|=|R(E(i_1))|=1,\ |R(E(i_2))|=2\ \text{or}\\|L(E(i_2))|=|R(E(i_2))|=1,\ |R(E(i_1))|=2\end{smallmatrix}\right\}}$$

$$+ \frac{1}{6}\mathbf{1}_{\left\{\begin{smallmatrix}|R(E(i_1))|=|R(E(i_2))|=2\\\nexists\ \min(i_1,i_2)\leq\ell<\max(i_1,i_2)\\\text{s.t.}\ |R(E(\ell))|=1\end{smallmatrix}\right\}} + \frac{1}{36}\mathbf{1}_{\left\{\begin{smallmatrix}E(i_1),E(i_2)\in U\\\exists\ \min(i_1,i_2)\leq\ell<\max(i_1,i_2)\\\text{s.t.}\ |R(E(\ell))|=1\end{smallmatrix}\right\}}. \tag{34}$$

Finally, for $E \in \Sigma^4_{a,even}(\vec{n})$, we will define

$$\#\text{loops}(E) = \#\{i \in [d] \mid |L(E(i))| = 1,\ |R(E(i))| = 2\}. \tag{35}$$

That is, a loop is created at layer $i$ if the four edges in $E$ all begin at occupy the same vertex in layer $i-1$ but occupy two different vertices in layer $i$. We have the following Lemma.

**Lemma 8** (Evaluation of Counting Terms in (30) and (31)). *Suppose $E \in \Sigma^4_{a_j,even}$ for some $j = 1, 2, 3, 4$. For each $i_1, i_2 \in \{1, \ldots, d\}$,*

$$\#\left\{\Gamma = (\gamma_1, \ldots, \gamma_4) \in \Gamma^4_{a_j,even} \mid \begin{smallmatrix}E^\Gamma=E,\ \Gamma(0)=a_j,\ t=1,2\\\gamma_1(i_t-1)=\gamma_2(i_t-1),\ \gamma_1(i_t)=\gamma_2(i_t)\end{smallmatrix}\right\}$$

*equals*

$$6^{\#loops(E)} A(E, i_1, i_2) \cdot \begin{cases}1, & j = 1, 2\\\widehat{C}(E, i_1, i_2), & j = 3, 4\end{cases}. \tag{36}$$

*Similarly,*

$$\#\left\{\Gamma \in \Gamma^4_{a_j,even} \mid \begin{smallmatrix}E^\Gamma=E,\ \Gamma(0)=a_j,\ t=1,2\\\gamma_1(i_1-1)=\gamma_2(i_1-1),\ \gamma_1(i_1)=\gamma_2(i_1)\\\gamma_2(i_2-1)=\gamma_3(i_2-1),\ \gamma_2(i_2)=\gamma_3(i_2)\end{smallmatrix}\right\}$$

*equals*

$$6^{\#loops(E)} A(E, i_1, i_2) C(E, i_1, i_2) \cdot \begin{cases}1, & j = 1, 2\\\widehat{C}(E, i_1, i_2), & j = 3, 4\end{cases}. \tag{37}$$

We prove Lemma 8 in §B.3 below. Assuming it for now, observe that

$$\frac{1}{36} \leq A(E, i_1, i_2) \leq 1$$

and that the conditions $L(E(1)) = a_j$ are the same for $j = 2, 3, 4$ since the equality it is in the sense of unordered multi-sets. Thus, we find that $\mathbb{E}[K^2_\text{w}]$ is bounded above/below by a constant times

$$\|x\|_4^4 \sum_{i_1,i_2=1}^d \sum_{E\in\Sigma^4_{a_1,even}} F_*(E) + (\|x\|_2^4 - \|x\|_4^4) \sum_{E\in\Sigma^4_{a_2,even}} F_*(E)(1 + 2\widehat{C}(E, i_1, i_2)). \tag{38}$$

Similarly, $\mathbb{E}[\Delta_\text{ww}]$ is bounded above/below by a constant times

$$\sum_{\substack{i_1,i_2=1\\i_1\neq i_2}}^d \left[\|x\|_4^4 \sum_{E\in\Sigma^4_{a_1,even}} F_*(E)6^{\#\text{loops}(E)}C(E, i_1, i_2)\right. \tag{39}$$

$$\left. + (\|x\|_2^4 - \|x\|_4^4) \sum_{E\in\Sigma^4_{a_2,even}} F_*(E)6^{\#\text{loops}(E)}C(E, i_1, i_2)(1 + 2\widehat{C}(E, i_2, i_2))\right].$$

Observe that every unordered multi-set four edge multiset $E \in \Sigma^4_{even}$ can be obtained by starting from some $V \in \Gamma^2$, considering its unordered edge multi-set $E^V$ and doubling all its edges. This map from $\Gamma^2$ to $\Sigma^4_{even}$ is surjective but not injective. The sizes of the fibers is computed by the following Lemma.

**Lemma 9.** *Fix $E \in \Sigma_{even}^4$. The number of $V \in \Gamma_Z^2$ so that $E = 2 \cdot E^V$ is $2^{\#loops(V) + \mathbf{1}_{\{|V(0)|=2\}}}$, where as in (35),*

$$\#loops(V) = \#\{i \in [d] \mid |V(i-1)| = 1, |V(i)| = 2\}.$$

Lemma 9 is proved in §B.4. Using it and that $0 \leq \widehat{C}(E, i_1, i_2) \leq 1$, the relation (38) shows that $\mathbb{E}[K_{\mathrm{w}}^2]$ is bounded above/below by a constant times

$$d^2 \sum_{V \in \Gamma^2} F_*(V) 3^{\#\mathrm{loops}(V)} \left( \|x\|_4^4 \mathbf{1}_{\{|V(0)|=1\}} + (\|x\|_2^4 - \|x\|_4^4) \mathbf{1}_{\{|V(0)|=2\}} \right). \quad (40)$$

Similarly, $\mathbb{E}[\Delta_{\mathrm{ww}}]$ is bounded above/below by a constant times

$$\sum_{\substack{i_1, i_2 = 1 \\ i_1 \neq i_2}}^{d} \sum_{V \in \Gamma^2} F_*(V) 3^{\#\mathrm{loops}(V)} C(V, i_1, i_2) \left( \|x\|_4^4 \mathbf{1}_{\{|V(0)|=1\}} + (\|x\|_2^4 - \|x\|_4^4) \mathbf{1}_{\{|V(0)|=2\}} \right), \quad (41)$$

where, in analogy to (32), we have

$$C(V, i_1, i_2) := \mathbf{1}_{\{\exists \ell = i_1, \dots, i_2 - 1 \text{ s.t. } |V(\ell)|=1\}}.$$

Let us introduce

$$\widehat{F}_*(V) := F_*(V) \cdot 3^{\#\mathrm{loops}(V)} \prod_{i=0}^{d} n_i^2$$

$$= 2^{\#\{i \in [d] \mid |V(i)|=1\}} 3^{\#\mathrm{loops}(V)} \mu_4^{\#\{i \in [d] \mid |V(i-1)|=|V(i)|=1\}}.$$

Since the number of $V$ in $\Gamma^2(\vec{n})$ with specified $V(0)$ equals $\prod_{i=1}^{d} n_i^2$, we find that so that for each $x \neq 0$, we have

$$\frac{\mathbb{E}[K_{\mathrm{w}}^2]}{\|x\|_2^4} \simeq \frac{d^2}{n_0^2} \mathcal{E}_x \left[ \widehat{F}_*(V) \right], \quad (42)$$

and similarly,

$$\frac{\mathbb{E}[\Delta_{\mathrm{ww}}]}{\|x\|_2^4} \simeq \frac{1}{n_0^2} \sum_{\substack{i_1, i_2 = 1 \\ i_1 \neq i_2}}^{d} \mathcal{E}_x \left[ \widehat{F}_*(V) C(V, i_1, i_2) \right].$$

Here, $\mathcal{E}_x$ is the expectation with respect to the probability measure on $V = (v_1, v_2) \in \Gamma^2$ obtained by taking $v_1, v_2$ independent, each drawn from the products of the measure $\left( x_1^2 / \|x\|_2^2, \dots, x_{n_0}^2 / \|x\|_2^2 \right)$ on $[n_0]$ and the uniform measure on $[n_i]$, $i = 1, \dots, d$.

We are now in a position to complete the proof of Theorems 1 and 2. To do this, we will evaluate the expectations $\mathcal{E}_x$ above to leading order in $\sum_i 1/n_i$ with the help of the following elementary result which is proven as Lemma 18 in Hanin & Nica (2018).

**Proposition 10.** *Let $A_0, A_1, \dots, A_d$ be independent events with probabilities $p_0, \dots, p_d$ and $B_0, \dots, B_d$ be independent events with probabilities $q_0, \dots, q_d$ such that*

$$A_j \cap B_j = \emptyset, \qquad \forall j = 0, \dots, d.$$

*Denote by $X_i$ the indicator that the event $A_i$ happens, $X_i := \mathbf{1}_{\{A_i\}}$, and by $Y_i$ the indicator that $B_i$ happens, $Y_i = \mathbf{1}_{\{B_i\}}$. Further, fix for every $i \in 1, \dots, d$ some $\alpha_i \geq 1, K_i \geq 1$ as well as $\gamma_i > 0$. Define*

$$Z = \prod_{i=1}^{d} \alpha_i^{X_i} \gamma_i^{X_{i-1} X_i} K_i^{Y_i}.$$

*Then, if $\gamma_i \geq 1$ for every $i$, we have:*

$$\mathbb{E}[Z] \leq \prod_{i=1}^{d} \left( 1 + p_i(\alpha_i - 1) + q_i(K_i - 1) + p_i p_{i-1} \alpha_i \alpha_{i-1} \gamma_{i-1}(\gamma_i - 1) \right), \quad (43)$$

*where by convention $\alpha_0 = \gamma_0 = 1$. In contrast, if $\gamma_i \leq 1$ for every $i$, we have:*

$$\mathbb{E}[Z] \geq \prod_{i=1}^{d} \left( 1 + p_i(\alpha_i - 1) + p_i p_{i-1} \alpha_{i-1} \alpha_i (\gamma_i - 1) \right) \quad (44)$$

We first apply Proposition 10 to the estimates above for $\mathbb{E}[K_{\mathrm{w}}^2]$. To do this, recall that

$$3^{\#\mathrm{loops}(V)} \;=\; \prod_{i=1}^{d} 3^{\mathbf{1}_{\{|V(i-1)|=1,\,|V(i)|=2\}}}.$$

Since $|V(d)| = 1$, we may also write

$$3^{\#\mathrm{loops}(V)} \;=\; \frac{1}{3}\prod_{i=1}^{d} 3^{\mathbf{1}_{\{|V(i-1)|=2,\,|V(i)|=1\}}} \;=\; \frac{1}{3}\prod_{i=1}^{d}\left(\frac{1}{3}\right)^{\mathbf{1}_{\{|V(i-1)|=|V(i)|=1\}}} 3^{\mathbf{1}_{\{|V(i)|=1\}}}.$$

Putting this together with (42) and noting that

$$\prod_{i=1}^{d} 2^{2-|V(i)|} = \prod_{i=1}^{d} 2^{\mathbf{1}_{\{|V(i)|=1\}}},$$

we find that

$$\mathbb{E}[K_{\mathrm{w}}^2]/\,\|x\|_2^4 \;\simeq\; \frac{1}{n_0^2}\mathcal{E}_x\left[\prod_{i=1}^{d}\left(\frac{\mu_4}{3}\right)^{\mathbf{1}_{\{|V(i-1)|=|V(i)|=1\}}} 6^{\mathbf{1}_{\{|V(i)|=1\}}}\right].$$

Since the contribution for each layer in the product is bounded above and below by constants, we have that $\mathbb{E}[K_{\mathrm{w}}^2]/\,\|x\|_2^4$ is bounded below by a constant times

$$\frac{d^2}{n_0^2}\mathcal{E}_x\left[\prod_{i=2}^{d-1}\left(1\wedge\frac{\mu_4}{3}\right)^{\mathbf{1}_{\{|V(i-1)|=|V(i)|=1\}}} 6^{\mathbf{1}_{\{|V(i)|=1\}}}\right] \tag{45}$$

and above by a constant times

$$\frac{d^2}{n_0^2}\mathcal{E}_x\left[\prod_{i=2}^{d-1}\left(1\vee\frac{\mu_4}{3}\right)^{\mathbf{1}_{\{|V(i-1)|=|V(i)|=1\}}} 6^{\mathbf{1}_{\{|V(i)|=1\}}}\right]. \tag{46}$$

Here, note that the initial condition given by $x$ and the terminal condition that all paths end at one neuron in the final layer are irrelevant. The expression (45) is there precisely $\mathbb{E}[Z_{d-1}/n_0^2]$ from Proposition 10 where $X_i$ is the event that $|V(i)| = 1$, $Y_i = \emptyset$, $\alpha_i = 6$, $\gamma_i = 1 \wedge \frac{\mu_4}{3} \le 1$, and $K_i = 1$. Thus, since for $i = 1, \ldots, d-1$, the probability of $X_i$ is $1/n_i + O(1/n_i^2)$, we find that

$$\mathbb{E}[K_{\mathrm{w}}^2]/\,\|x\|_2^4 \;\ge\; \frac{d^2}{n_0^2}\prod_{i=2}^{d-1}\left(1 + \frac{5}{n_i} + O\left(\frac{1}{n_i^2}+\frac{1}{n_{i-1}^2}\right)\right) \ge \frac{d^2}{n_0^2}\exp\left(5\sum_{i=2}^{d-1}\frac{1}{n_i} + O\left(\sum_{i=2}^{d-1}\frac{1}{n_i^2}\right)\right),$$

where in the last inequality we used that $1 + x \ge e^{x - x^2/2}$ for $x \ge 0$. Since $e^{-1/n_1 + 1/n_d} \simeq 1$, we conclude

$$\mathbb{E}[K_{\mathrm{w}}^2]/\,\|x\|_2^4 \;\ge\; \frac{d^2}{n_0^2}\exp\left(5\beta\right)\left(1 + O\left(\beta^{-1}\sum_{i=1}^{d}\frac{1}{n_i^2}\right)\right), \qquad \beta = \sum_{i=1}^{d}\frac{1}{n_i}.$$

When combined with (23) this gives the lower bound in Proposition 3. The matching upper bound is obtained from (46) in the same way using the opposite inequality from Proposition 10.

To complete the proof of Proposition 3, we prove the analogous bounds for $\mathbb{E}[\Delta_{\mathrm{ww}}]$ in a similar fashion. Namely, we fix $1 \le i_1 < i_2 \le d$ and write

$$C(V, i_1, i_2) \;=\; \sum_{\ell=i_1}^{i_2-1}\mathbf{1}_{A_\ell}, \qquad A_\ell \;:=\; \left\{\begin{array}{c}|V(i)|=2,\,i=i_1,\ldots,\ell-1 \\ \text{and } |V(\ell)|=1\end{array}\right\}.$$

The set $A_\ell$ is the event that the first collision between layers $i_1, i_2$ occurs at layer $\ell$. We then have

$$\mathcal{E}_x\left[\widehat{F}_*(V)C(V, i_1, i_2)\right] \;=\; \sum_{\ell=i_1}^{i_2-1}\mathcal{E}_x\left[\widehat{F}_*(V)\mathbf{1}_{\{A_\ell\}}\right],$$

On the event $A_\ell$, notice that $\widehat{F}_*(V)$ only depends on the layers $1 \leq i \leq i_1$ and layers $\ell < i \leq d$ because the event $A_\ell$ fixes what happens in layers $i_1 < i \leq \ell$. Mimicking the estimates (45), (46) and the application of Proposition 10 and using independence, we get that:

$$\mathcal{E}_x \left[ \widehat{F}_*(V) 1\{A_\ell\} \right] \simeq \exp \left( \sum_{\substack{i=1 \\ i \notin [i_1, \ell)}}^d \frac{1}{n_i} \right) \mathcal{E}_x \left( \mathbf{1}_{\{A_\ell\}} \right)$$

Finally, we compute:

$$\mathcal{E}_x \left( \mathbf{1}_{\{A_\ell\}} \right) = \mathbb{P}(A_\ell) = = \frac{1}{n_\ell} \prod_{i=i_1}^{\ell-1} \left( 1 - \frac{1}{n_i} \right) \simeq \frac{1}{n_\ell} \exp \left( -\sum_{i=i_1}^{\ell-1} \frac{1}{n_i} \right),$$

Combining this we obtain that $\mathbb{E}[\Delta_{\mathrm{ww}}] / \|x\|_2^4$ is bounded above and below by constants times

$$\frac{1}{n_0^2} \left[ \sum_{\substack{i_1, i_2=1 \\ i_i < i_2}}^d \sum_{\ell=i_1}^{i_2-1} \frac{1}{n_\ell} e^{-5/n_\ell - 6 \sum_{i=i_1}^{\ell-1} \frac{1}{n_i}} \right] \exp \left( 5 \sum_{i=1}^d \frac{1}{n_i} \right) \left( 1 + O \left( \sum_{i=1}^d \frac{1}{n_i^2} \right) \right).$$

This completes the proof of Proposition 3, modulo the proofs of Lemmas 6-9, which we supply below. $\qquad \square$

### B.1 Proof of Lemma 6

Fix an input $x \in \mathbb{R}^{n_0}$ to $\boldsymbol{N}$. We will continue to write as in (2) $y^{(i)}$ for the vector of pre-activations as layer $i$ corresponding to $x$. We need the following simple Lemma.

**Lemma 11.** *With probability* $1$, *either there exists* $i$ *so that* $y^{(i)} = 0$ *or, for every* $i \in [d], j \in [n_i]$ *we have* $y_j^{(i)} \neq 0$.

*Proof.* The argument is similar to Lemma 8 in Hanin & Rolnick (2019). Namely, fix $i \in [d], j \in [n_i]$. If $y^{(\ell)} \neq 0$ for every $\ell$, then there exists at least one path $\gamma$ in the computational graph of the map $x \mapsto y_j^{(i)}$ so that, $y_\gamma^{(\ell)} > 0$ for each $\ell = 1, \ldots, i-1$. For event that $y_j^{(i)} = 0$ is therefore contained in the union over all *non-empty* subsets $\Gamma$ of the collection of all paths in the computational graph of $x \mapsto y_j^{(i)}$ of the event that

$$\sum_{\gamma \in \Gamma} \prod_{\ell=1}^i \widehat{W}_\gamma^{(\ell)} = 0.$$

For each fixed $\Gamma$ this event defines a co-dimension $1$ set in the space of all the weights. Hence, since the joint distribution of the weights has a density with respect to Lebesgue measure (see just before (4)), the union of this (finite number) of events has measure $0$. This shows that on the even that $y^{(\ell)} \neq 0$ for every $\ell$, $y_j^{(i)} \neq 0$ with probability $1$. Taking the union over $i, j$ completes the proof. $\quad \square$

Lemma 11 shows that for our fixed $x$, with probability $1$, the derivative of each $\xi_j^{(i)}$ in (19) vanishes. Hence, almost surely, for any edge $e$ in the computational graph of $\boldsymbol{N}$:

$$\frac{\partial \boldsymbol{N}}{\partial W_e^{(j)}}(x) = \sum_{a=1}^{n_0} x_a \sum_{\substack{\gamma \in \Gamma_a^1 \\ e \in \gamma}} \frac{\mathrm{wt}(\gamma)}{W_e}. \tag{47}$$

This proves the formulas for $K_{\boldsymbol{N}}, K_{\boldsymbol{N}}^2$. To derive the result for $\Delta K_{\boldsymbol{N}}$, we write

$$\Delta K_{\boldsymbol{N}} = -\lambda \sum_{\text{edges } e} \left( \frac{\partial}{\partial W_e} K_{\boldsymbol{N}} \right) \frac{\partial \mathcal{L}}{\partial W_e},$$

where the loss $\mathcal{L}$ on a single batch containing only $x$ is $\frac{1}{2}\left(\boldsymbol{N}(x) - \boldsymbol{N}_*(x)\right)^2$. We therefore find

$$\Delta K_{\boldsymbol{N}} = -2\lambda \sum_{\text{edges } e_1, e_2} \frac{\partial \boldsymbol{N}}{\partial W_{e_1}} \frac{\partial^2 \boldsymbol{N}}{\partial W_{e_1} \partial W_{e_2}} \frac{\partial \boldsymbol{N}}{\partial W_{e_2}} \left(\boldsymbol{N}(x) - \boldsymbol{N}_*(x)\right).$$

Using (47) and again applying Lemma 11, we find that with probability 1

$$\frac{\partial^2 \boldsymbol{N}}{\partial W_{e_1} \partial W_{e_2}} = \sum_{a=1}^{n_0} x_a \sum_{\substack{\gamma \in \Gamma_a^1 \\ e_1, e_2 \in \gamma, \, e_1 \neq e_2}} \frac{\text{wt}(\gamma_1)\text{wt}(\gamma_2)}{W_{e_1} W_{e_2}}.$$

Thus, almost surely

$$-\frac{1}{2\lambda} \Delta K_{\boldsymbol{N}} = \sum_{a \in [n_0]^4} \prod_{k=1}^{4} x_{a_k} \sum_{\Gamma_a \in \Gamma^4(\vec{n})} \sum_{\substack{e_1 \in \gamma_1, \gamma_2 \\ e_2 \in \gamma_2, \gamma_3 \\ e_1 \neq e_2}} \frac{\prod_{k=1}^{4} \text{wt}(\gamma_k)}{W_{e_1}^2 W_{e_2}^2}$$

$$- \boldsymbol{N}_*(x) \sum_{a \in [n_0]^3} \prod_{k=1}^{3} x_{a_k} \sum_{\Gamma \in \Gamma_a^3(\vec{n})} \sum_{\substack{e_1 \in \gamma_1, \gamma_2 \\ e_2 \in \gamma_2, \gamma_3 \\ e_1 \neq e_2}} \frac{\prod_{k=1}^{4} \text{wt}(\gamma_k)}{W_{e_1}^2 W_{e_2}^2}.$$

To complete the proof of Lemma 6 it therefore remains to check that this last term has mean $0$. To do this, recall that the output layer of $\boldsymbol{N}$ is assumed to be linear and that the distribution of each weight is symmetric around $0$ (and hence has vanishing odd moments). Thus, the expectation over the weights in layer $d$ has either 1 or 3 weights in it and so vanishes. □

## B.2 Proof of Lemma 7

Lemma 7 is almost a corollary of of Theorem 3 in Hanin (2018) and Proposition 2 in Hanin & Nica (2018). The difference is that, in Hanin (2018); Hanin & Nica (2018), the biases in $\boldsymbol{N}$ were assumed to have a non-degenerate distribution, whereas here we've set them to zero. The non-degeneracy assumption is not really necessary, so we repeat here the proof from Hanin (2018) with the necessary modifications.

If $x = 0$, then $\mathcal{N}(x) = 0$ for any configuration of weights since the network biases all vanish. Will therefore suppose that $x \neq 0$. Let us first show (20). We have from Lemma 6 that

$$\mathbb{E}[K_{\boldsymbol{N}}(x, x)] = \sum_{a \in [n_0]^2} x_{a_1} x_{a_2} \sum_{\Gamma \in \Gamma_{a, even}^2} \sum_{e \in \gamma_1 \cap \gamma_2} \mathbb{E}\left[\frac{\prod_{k=1}^{2} \text{wt}(\gamma_k)}{W_e^2}\right]. \tag{48}$$

To compute the inner expectation, write $\boldsymbol{F}_j$ for the sigma algebra generated by the weight in layers up to and including $j$. Let us also define the events:

$$S_j := \{x^{(j)} \neq 0\},$$

where we recall from (2) that $x^{(j)}$ are the post-activations in layer $j$. Supposing first that $e$ is not in layer $d$, the expectation becomes

$$\mathbb{E}\left[\frac{\prod_{i=1}^{d-1} \widehat{W}_{\gamma_1}^{(i)} \widehat{W}_{\gamma_2}^{(i)} \mathbf{1}_{\{y_{\gamma_1}^{(i)} > 0\}} \mathbf{1}_{\{y_{\gamma_2}^{(i)} > 0\}}}{W_e^2} \mathbb{E}\left[\widehat{W}_{\gamma_1}^{(d)} \widehat{W}_{\gamma_2}^{(d)} \,\middle|\, \mathcal{F}_{d-1}\right]\right].$$

We have

$$\mathbb{E}\left[\widehat{W}_{\gamma_1}^{(d)} \widehat{W}_{\gamma_2}^{(d)} \,\middle|\, \mathcal{F}_{d-1}\right] = \frac{1}{n_{d-1}} \mathbf{1}_{\left\{\substack{\gamma_1(d-1)=\gamma_2(d-1) \\ \gamma_1(d)=\gamma_2(d)}\right\}}$$

Thus, the expectation in (48) becomes $\frac{1}{n_{d-1}} \mathbf{1}_{\left\{\substack{\gamma_1(d-1)=\gamma_2(d-1) \\ \gamma_1(d)=\gamma_2(d)}\right\}}$ times

$$\mathbb{E}\left[\frac{\prod_{k=1}^{2} \prod_{i=1}^{d-2} \widehat{W}_{\gamma_k}^{(i)} \mathbf{1}_{\{y_{\gamma_k}^{(i)} > 0\}}}{W_e^2} \mathbb{E}\left[\prod_{k=1}^{2} \widehat{W}_{\gamma_k}^{(d-1)} \mathbf{1}_{\{y_{\gamma_k}^{(d-1)} > 0\}} \,\middle|\, \mathcal{F}_{d-2}\right]\right].$$

Note that given $\mathcal{F}_{d-2}$, the pre-activations $y_j^{(d-1)}$ of different neurons in layer $d-1$ are independent. Hence,

$$
\mathbb{E}\left[\prod_{k=1}^{2}\widehat{W}_{\gamma_k}^{(d-1)}\mathbf{1}_{\{y_{\gamma_k}^{(d-1)}>0\}}\,\bigg|\,\mathcal{F}_{d-2}\right] = \begin{cases} \prod_{k=1}^{2}\mathbb{E}\left[\widehat{W}_{\gamma_k}^{(d-1)}\mathbf{1}_{\{y_{\gamma_k}^{(d-1)}>0\}}\,\bigg|\,\mathcal{F}_{d-2}\right], & \gamma_1(d-1)\neq\gamma_2(d-1) \\ \mathbb{E}\left[\mathbf{1}_{\{y_{\gamma_1}^{(d-1)}>0\}}\prod_{k=1}^{2}\widehat{W}_{\gamma_k}^{(d-1)}\,\bigg|\,\mathcal{F}_{d-2}\right], & \gamma_1(d-1)=\gamma_2(d-1) \end{cases}.
$$

Recall that by assumption, the weight matrix $\widehat{W}^{(d-1)}$ in layer $d-1$ is equal in distribution to $-\widehat{W}^{(d-1)}$. This replacement leaves the product $\prod_{k=1}^{2}\widehat{W}_{\gamma_k}^{(d-1)}$ unchanged but changes $\mathbf{1}_{\{y_{\gamma_1}^{(d-1)}>0\}}$ to $\mathbf{1}_{\{y_{\gamma_1}^{(d-1)}\leq 0\}}$. On the event $S_{d-1}$ (which occurs whenever $y_{\gamma_k}^{(d-2)}>0$) we have that $y_{\gamma_1}^{(d-1)}\neq 0$ with probability 1 since we assumed that the distribution of each weight has a density relative to Lebesgue measure. Hence, symmetrizing over $\pm\widehat{W}^{(d)}$, we find that

$$
\mathbb{E}\left[\prod_{k=1}^{2}\widehat{W}_{\gamma_k}^{(d-1)}\mathbf{1}_{\{y_{\gamma_k}^{(d-1)}>0\}}\,\bigg|\,\mathcal{F}_{d-2}\right] = \frac{1}{n_{d-2}}\mathbf{1}_{\left\{\substack{\gamma_1(d-1)=\gamma_2(d-1)\\\gamma_1(d-2)=\gamma_2(d-2)}\right\}}.
$$

Similarly, if $e$ is in layer $i$, then we automatically find that $\gamma_1(i-1)=\gamma_2(i-1)$ and $\gamma_1(i)=\gamma_2(i)$, giving an expectation of $1/n_{i-1}\mathbf{1}_{\left\{\substack{\gamma_1(i)=\gamma_2(i)\\\gamma_1(i-1)=\gamma_2(i-1)}\right\}}$. Proceeding in this way yields

$$
\mathbb{E}[K_{\boldsymbol{N}}(x,x)] = \sum_{a\in[n_0]^2}x_{a_1}x_{a_2}\prod_{i=1}^{d}\frac{1}{n_{i-1}}\sum_{\Gamma\in\Gamma_{a,even}^2(\vec{n}}\sum_{e\in\gamma_1\cap\gamma_2}\delta_{\gamma_1=\gamma_2}
$$

$$
= \sum_{a\in[n_0]^2}x_{a_1}x_{a_2}\sum_{\Gamma\in\Gamma_{a,even}^2(\vec{n})}\delta_{\gamma_1=\gamma_2}\prod_{i=1}^{d}\frac{1}{n_{i-1}},
$$

which is precisely (20). The proofs of (21) and (22) are similar. We have

$$
\mathbb{E}[K_{\boldsymbol{N}}(x,x)^2] = \sum_{a\in[n_0]^4}\prod_{k=1}^{4}x_{a_k}\sum_{\Gamma\in\Gamma_a^4}\sum_{\substack{e_1\in\gamma_1,\gamma_2\\e_2\in\gamma_3,\gamma_4}}\mathbb{E}\left[\frac{\prod_{k=1}^{4}\mathrm{wt}(\gamma_k)}{W_{e_1}^2W_{e_2}^2}\right].
$$

As before let us first assume that edges $e_1,e_2$ are not in layer $d$. Then,

$$
\mathbb{E}\left[\prod_{k=1}^{4}\mathrm{wt}(\gamma_k)\right] = \mathbb{E}\left[\prod_{k=1}^{4}\prod_{i=1}^{d-1}\widehat{W}_{\gamma_k}^{(i)}\mathbf{1}_{\{y_{\gamma_k}^{(i)}>0\}}\mathbb{E}\left[\prod_{k=1}^{4}\widehat{W}_{\gamma_k}^{(d)}\,\bigg|\,\mathcal{F}_{d-1}\right]\right].
$$

The the inner expectation is

$$
\mathbf{1}_{\left\{\substack{\text{each weight appears an}\\\text{even number of times}}\right\}}\cdot\frac{1}{n_{d-1}^2}\mu_4^{\mathbf{1}_{\{|\Gamma(d-1)|=|\Gamma(d)|=1\}}}.
$$

In contrast, if $d=\ell(e_1)$ or $d=\ell(e_2)$, then the inner expectation is

$$
\mathbf{1}_{\left\{\substack{\text{each weight appears an}\\\text{even number of times}}\right\}}\frac{1}{n_{d-1}^2}.
$$

Again symmetrizing with respect to $\pm\widehat{W}^{(d)}$ and using that the pre-activation of different neurons are independent given the activations in the previous layer we find that, on the event $\{y_{\gamma_k}^{(d-2)}>0\}$,

$$
\mathbb{E}\left[\prod_{k=1}^{4}\widehat{W}_{\gamma_k}^{(d-1)}\mathbf{1}_{\{y_{\gamma_k}^{(d-1)}>0\}}\,\bigg|\,\mathcal{F}_{d-2}\right] = \mathbf{1}_{\left\{\substack{\text{each weight appears an}\\\text{even number of times}}\right\}}\frac{2^{2-|\Gamma(d-1)|}}{n_{d-1}^2}\mu_4^{\mathbf{1}_L},
$$

where $L$ is the event that $|\Gamma(d-1)|=|\Gamma(d)|=1$ and $e_1,e_2$ are not in layer $d-1$. Proceeding in this way one layer at a time completes the proofs of (21) and (22). $\qquad\square$

### B.3 PROOF OF LEMMA 8

Fix $j = 1, \ldots, 4$, edges $e_1, e_2$ with $\ell(e_1) \leq \ell(e_2)$ in the computational graph of $N$ and $E \in \Sigma^4_{a_j, even}$. The key idea is to decompose $E$ into loops. To do this, define

$$i_0 = -1, \qquad i_k(E) := \min\{i > i_{k-1} \mid |L(E(i))| = 1, |R(E(i))| = 2\}, \; k \geq 1, \ldots, \#\text{loops}(E).$$

For each $i = 1, \ldots, d$ there exists unique $k = 1, \ldots, \#\text{loops}(E)$ so that

$$i_{k-1}(E) \leq i < i_k(E).$$

We will say that two layers $i, j = 1, \ldots, d$ belong to the same loop of $E$ if exists $k = 1, \ldots, \#\text{loops}(E)$ so that

$$i_{k-1}(E) \leq i, j < i_k(E).$$

We proceed layer by layer to count the number of $\Gamma \in \Gamma^4_{a_j, even}$ satisfying $\Gamma(0) = a_j$ and $E^\Gamma = E$. To do this, suppose we are given $\Gamma(i-1) \in [n_{i-1}]^4$ and we have $L(E(i)) = 2$. Then $\Gamma(i-1)$ is some permutation of $(\alpha_1, \alpha_1, \alpha_2, \alpha_2)$ with $\alpha_1 \neq \alpha_2$. Moreover, for $j = 1, 2$ there is a unique edge (with multiplicity 2) in $E(i)$ whose left endpoint is $\alpha_j$. Therefore, $\Gamma(i-1)$ determines $\Gamma(i)$ when $L(E(i)) = 2$. In contrast, suppose $L(E(i)) = 1$. If $R(E(i)) = 1$, then $E(i)$ consists of a single edge with multiplicity 4, which again determines $\Gamma(i-1), \Gamma(i)$. In short, $\Gamma(i)$ determines $\Gamma(j)$ for all $j$ belonging to the same loop of $E$ as $i$. Therefore, the initial condition $\Gamma(0) = a_j$ determines $\Gamma(i)$ for all $i \leq i_1$ and the conditions $e_1 \in \gamma_1, e_2 \in \gamma_2$ determine $\Gamma$ in the loops of $E$ containing the layers of $e_1, e_2$.

Finally, suppose $L(E(i)) = 1$ and $R(E(i)) = 2$ (i.e. $i = i_k(E)$ for some $k = 1, \ldots, d$) and that $e_1, e_2$ are not contained in the same loop of $E$ layer $i$. Then all $\binom{4}{2} = 6$ choices of $\Gamma(i)$ satisfy $\Gamma(i) = R(E(i))$, accounting for the factor of $6^{\#\text{loops}(E)}$. The concludes the proof in the case $j = 1$. the only difference in the cases $j = 2, 3, 4$ is that if $\gamma_1(0) \neq \gamma_2(0)$ (and hence $\gamma_3(0) \neq \gamma_4(0)$), then since $\ell(e_1) \leq \ell(e_2)$ in order to satisfy $e_1 \in \gamma_1, \gamma_2$ we must have that $i_1(E) < \ell(e_1)$. $\qquad\square$

### B.4 PROOF OF LEMMA 9

The proof of Lemma 9 is essentially identical to the proof of Lemma 8. In fact it is slightly simpler since there are no distinguished edges $e_1, e_2$ to consider. We omit the details. $\qquad\square$

## C PROOF OF PROPOSITION 4

In this section, we seek to estimate $\mathbb{E}[K_b], \mathbb{E}[K_b^2], \mathbb{E}[\Delta_{bb}]$. The approach is essentially identical to but somewhat simpler than our proof of Proposition 3 in §B. We will therefore focus here on explaining the salient differences. Our starting point is the following analog of Lemma 6, which gives a sum-over-paths expression for the bias contribution $K_b$ to the neural tangent kernel. To state it, let us define, for any collection $Z = (z_1, \ldots, z_k) \in Z^k$ of $k$ neurons in $N$

$$\mathbf{1}_{\{y_Z > 0\}} := \prod_{j=1}^{k} \mathbf{1}_{\{y_{z_j} > 0\}},$$

to be the event that the pre-activations of the neurons $z_k$ are positive.

**Lemma 12** ($K_b$ as a sum over paths). *With probability* 1,

$$K_b = \sum_{Z \in Z^1} \mathbf{1}_{\{Z > 0\}} \sum_{\Gamma \in \Gamma^2_{(Z,Z)}} \prod_{k=1}^{2} \text{wt}(\gamma_k), \tag{49}$$

*where $Z^1$, $\Gamma^2_{(Z,Z)}$, $\text{wt}(\gamma)$ are defined in §A. Further, almost surely,*

$$\Delta_{bb} = 0. \tag{50}$$

The proof of this result is a small modification of the proof of Lemma 6 and hence is omitted. Taking expectations, we therefore obtain the following analog to Lemma 7.

**Lemma 13** (Expectation of $K_{\mathrm{b}}, K_{\mathrm{b}}^2$ as a sum over paths). *We have*

$$\mathbb{E}[K_{\mathrm{b}}] = \frac{1}{2} \sum_{Z \in Z^1} \sum_{\substack{\Gamma \in \Gamma^2_{(Z,Z),even} \\ \Gamma=(\gamma_1,\gamma_2)}} H(\Gamma), \qquad H(\Gamma) = \mathbf{1}_{\{\gamma_1=\gamma_2\}} \prod_{i=\ell(Z)+1}^{d} \frac{1}{n_{i-1}}. \qquad (51)$$

*Moreover,*

$$\mathbb{E}[K_{\mathrm{b}}^2] = \frac{1}{2} \sum_{\substack{Z=(z_1,z_2)\in Z^2 \\ \ell(z_1)\leq\ell(z_2)}} \sum_{\Gamma \in \Gamma^4_{(Z,Z),even}} \widehat{H}(\Gamma),$$

*where for $\Gamma = (\gamma_1, \ldots, \gamma_4) \in \Gamma^4_{(Z,Z),even}$ we have*

$$\widehat{H}(\Gamma) = \prod_{i'=\ell(z_1)+1}^{\ell(z_2)} \mathbf{1}_{\left\{ \substack{\gamma_1(i')=\gamma_2(i') \\ \gamma_1(i'-1)=\gamma_2(i'-1)} \right\}} \frac{1}{n_{i'-1}} \prod_{i=\ell(z_2)+1}^{d} \frac{2^{|\Gamma(i)|-2}}{n_{i-1}^2} \mu_4^{\mathbf{1}_{\{|\Gamma(i)|=|\Gamma(i-1)|\}}}. \qquad (52)$$

The proof is identical to the argument used in §B.2 to establish Lemma 7, so we omit the details. The relation (51) is easy to simplify:

$$\mathbb{E}[K_{\mathrm{b}}] = \frac{1}{2} \sum_{Z \in Z^1} \sum_{\gamma \in \Gamma_Z} \prod_{i=\ell(Z)+1}^{d} \frac{1}{n_{i-1}} = \frac{1}{2} \sum_{z \in Z^1} \frac{1}{n_{\ell(z)}} = \frac{d}{2},$$

where we used that the number paths from a neuron in layer $\ell$ to the output of $N$ equals $\prod_{i=\ell+1}^{d} n_i$. This proves the first statement in Proposition 4. Next, let us explain how to simplify $\mathbb{E}[K_{\mathrm{b}}^2]$. The key computation is the following

**Lemma 14.** *Fix two neurons $z_1, z_2$ with $\ell(z_1) \leq \ell(z_2)$ and write $Z = (z_1, z_1, z_2, z_2)$. Then,*

$$\sum_{\Gamma \in \Gamma^4_{Z,even}} \widehat{H}(\Gamma) \simeq \frac{1}{n_{\ell(z_1)} n_{\ell(z_2)}} \exp\left(5 \sum_{i=\ell(z_2)+1}^{d} \frac{1}{n_i}\right) \left(1 + O\left(\sum_{i=\ell(z_2)+1}^{d} \frac{1}{n_i^2}\right)\right). \qquad (53)$$

*Proof.* The proof of Lemma 14 is a simplified version of the computation of $\mathbb{E}[K_{\mathrm{w}}^2]$ (starting around (24) and ending at the end of the proof of Proposition 3). Specifically, note that for $\Gamma = (\gamma_1, \ldots, \gamma_4) \in \Gamma^4_{Z,even}$ with $\ell(z_1) \leq \ell(z_2)$, the delta functions $\mathbf{1}_{\{\gamma_1(i')=\gamma_2(i')\}}\mathbf{1}_{\{\gamma_1(i'-1)=\gamma_2(i'-1)\}}$ in the definition (52) of $\widehat{H}(\Gamma)$ ensures that $\gamma_1, \gamma_2$ go through the same neuron in layer $\ell(z_2)$. To condition on the index of this neuron, we recall that we denote by $z(j, \beta)$ neuron number $\beta$ in layer $j$. We have

$$\sum_{\Gamma \in \Gamma^4_{Z,even}} \widehat{H}(\Gamma) = \sum_{\beta=1}^{n_{\ell(z_2)}} \sum_{\gamma_1,\gamma_2:z_1 \to z(\ell(z_2),\beta)} \prod_{i'=\ell(z_1)+1}^{\ell(z_2)} \mathbf{1}_{\left\{ \substack{\gamma_1(i')=\gamma_2(i') \\ \gamma_1(i'-1)=\gamma_2(i'-1)} \right\}} \frac{1}{n_{i'-1}} \sum_{\Gamma \in \Gamma^4_{Z',even}} H(\Gamma)$$

$$= \frac{1}{n_{\ell(z_1)-1}} \sum_{\beta=1}^{n_{\ell(z_2)}} \sum_{\Gamma \in \Gamma^4_{Z',even}} H(\Gamma), \qquad (54)$$

where $Z' = (z(\ell(z_2), \beta), z(\ell(z_2), \beta), z_2, z_2)$ and

$$H(\Gamma) = \prod_{i=\ell(z_2)+1}^{d} \frac{2^{|\Gamma(i)|-2}}{n_{i-1}^2} \mu_4^{\mathbf{1}_{\{|\Gamma(i)|=|\Gamma(i-1)|\}}}.$$

Since the inner sum in (54) is independent of $\beta$ by symmetry, we find

$$\sum_{\Gamma \in \Gamma^4_{Z,even}} \widehat{H}(\Gamma) = \frac{n_{\ell(z_2)}}{n_{\ell(z_1)-1}} \sum_{\Gamma \in \Gamma^4_{Z'',even}} H(\Gamma), \qquad (55)$$

where $Z'' = (1, 1, z_2, z_2)$. The inner sum in (55) is now precisely one of the terms $I_j$ from (27) without counting terms involving edges $e_1, e_2$, except that the paths start at neuron 1 in layer $\ell(z_2)$. The changes of variables from $\Gamma \in \Gamma_{even}^4$ to $E \in \Sigma_{even}^4$ to $V \in \Gamma^2$ that we used to estimate the $I_j$'s are no far simpler. In particular, Lemma 8 still holds but without any of the $A(E, i_1, i_2)$, $C(E, i_1, i_2), \widehat{C}(E, i_1, i_2)$ terms. Thus, we find that

$$\sum_{\Gamma \in \Gamma_{Z''}^{4, even}} H(\Gamma) \simeq \sum_{E \in \Sigma_{Z''}^{4, even}} H(E) 6^{\#\text{loops}(E)} \simeq \sum_{V \in \Gamma_{Z'''}^2} H(V) 3^{\#\text{loops}(V)},$$

where for the second estimate we applied Lemma 9 and have written $Z''' = (1, z_2)$. Thus, as in the derivation of (42), we find that

$$\sum_{\Gamma \in \Gamma_{Z''}^{4, even}} H(\Gamma) \simeq \frac{1}{n_{\ell(z_2)}^2} \mathcal{E}\left[H_*(V)\right],$$

where

$$H_*(V) = 2^{\#\{i \in [d] \mid |V(i)|=1\}} 3^{\#\text{loops}(V)} \mu_4^{\#\{i \in [d] \mid |V(i-1)|=|V(i)|=1\}}$$

and $\mathcal{E}$ is the expectation over pairs of paths starting from neurons $1, z_2$ in layer $\ell(z_2)$ to the output of the network for which neurons in subsequent layers are chosen independently and uniformly among all neurons in that layer. This is precisely the expectation we evaluated in the end of the proof for Proposition 3. Thus, applying Proposition 10 exactly as in that case, we find that

$$\sum_{\Gamma \in \Gamma_Z^{4, even}} F(\Gamma) \simeq \frac{1}{n_{\ell(z_2)}^2} \exp\left(5 \sum_{i=\ell(z_2)+1}^d \frac{1}{n_i} + O\left(\sum_{i=\ell(z_2)+1}^d \frac{1}{n_i^2}\right)\right).$$

Putting this together with (54) completes the proof of Lemma 14. $\qquad\square$

Lemma 14 combined with Lemma 13 yields

$$\mathbb{E}[K_b^2] \simeq \sum_{\substack{i,j=1 \\ i \leq j}}^d \exp\left(5 \sum_{i=j+1}^d \frac{1}{n_i}\right)\left(1 + O\left(\sum_{i=1}^d \frac{1}{n_i^2}\right)\right),$$

as claimed in the statement Proposition 4. $\qquad\square$

## D   PROOF OF PROPOSITION 5

We begin by computing $\mathbb{E}[K_b K_w]$. We will use a hybrid of the procedures for computing $\mathbb{E}[K_b^2]$ and $\mathbb{E}[K_w^2]$. Recall from Lemmas 6 and 12 that

$$K_b = \sum_{Z \in Z^1} \mathbf{1}_{\{yZ>0\}} \sum_{\Gamma \in \Gamma_{(Z,Z)}^2} \prod_{k=1}^2 \text{wt}(\gamma_k), \qquad K_w = \sum_{a \in [n_0]^2} \prod_{k=1}^2 x_{a_k} \sum_{\substack{\Gamma \in \Gamma_a^2 \\ \Gamma=(\gamma_1, \gamma_2)}} \sum_{e \in \gamma_1, \gamma_2} \frac{\prod_{k=1}^2 \text{wt}(\gamma_k)}{W_e^2}.$$

Therefore, the expectation of the product $\mathbb{E}[K_b K_w]$ has the following form

$$\sum_{a \in [n_0]^2} \prod_{k=1}^2 x_{a_k} \sum_{Z \in Z^1} \sum_{\Gamma \in \Gamma_{(Z,Z,a)}^4} \sum_{e \in \gamma_3, \gamma_4} \mathbb{E}\left[\mathbf{1}_{\{yZ>0\}} \frac{\prod_{k=1}^4 \text{wt}(\gamma_k)}{W_e^2}\right].$$

Here, for a neuron $Z$ and $a = (a_1, a_2) \in [n_0]^2$ we've denoted by $\Gamma_{(Z,Z,a)}^4$ the set of four tuples $(\gamma_1, \ldots, \gamma_4)$ of paths in the computational graph of $N$ where $\gamma_1, \gamma_2$ start from $Z$ and $\gamma_3, \gamma_4$ start at neurons $a_1, a_2$ respectively. The analog of Lemmas 7 and 13 (with essentially the same proof), gives that the expectation in the previous line equals

$$\frac{\|x\|^2}{2} \prod_{i=1}^{\ell(Z)} \mathbf{1}_{\{\gamma_3(i)=\gamma_4(i)\}} \frac{1}{n_{i-1}} \prod_{i=\ell(Z)+1}^d \frac{2^{2-|\Gamma(i)|}}{n_{i-1}^2} \mu_4^{\mathbf{1}_{\{|\Gamma(i-1)|=|\Gamma(i)|=1, \, i \neq \ell(e)\}}},$$

which, up to a multiplicative constant equals

$$G_Z(\Gamma) \ := \ \|x\|^2 \prod_{i=1}^{\ell(Z)} \mathbf{1}_{\{\gamma_3(i)=\gamma_4(i)\}} \frac{1}{n_{i-1}} \prod_{i=\ell(Z)+1}^{d} \frac{2^{2-|\Gamma(i)|}}{n_{i-1}^2} \mu_4^{\mathbf{1}_{\{|\Gamma(i-1)|=|\Gamma(i)|=1\}}}, \qquad (56)$$

which is independent of $e$. Thus, we find

$$\mathbb{E}[K_{\mathrm{b}}K_{\mathrm{w}}] \ \simeq \ \|x\|^2 \sum_{i=1}^{d} \sum_{Z \in Z^1} \sum_{\Gamma \in \Gamma_{(Z,Z,1,1),even}^4} G_Z(\Gamma) T_{3,4}^i(\Gamma),$$

where if $\Gamma = (\gamma_1, \ldots, \gamma_4)$ we recall that $T_{3,4}^i(\Gamma)$ is the indicator function of the event that paths $\gamma_3, \gamma_4$ pass through the same edge in the computational graph of $N$ at layer $i$ (see (29)).

As before, note that the delta functions $\mathbf{1}_{\{\gamma_3(i)=\gamma_4(i)\}}$ ensure that $\gamma_3, \gamma_4$ pass through the same neuron in layer $\ell(Z)$. Thus, we may condition on the common neuron through which $\gamma_3, \gamma_4$ must pass at layer $\ell(Z)$ to obtain that $\mathbb{E}[K_{\mathrm{b}}K_{\mathrm{w}}]$ is bounded above and below by a constant times

$$\|x\|^2 \sum_{i=1}^{d} \sum_{Z \in Z^1} \sum_{\Gamma \in \Gamma_{(Z,Z,1,1),even}^4} G_Z(\Gamma) T_{3,4}^i(\Gamma) \ = \ \frac{\|x\|^2}{n_0} \sum_{i=1}^{d} \sum_{Z \in Z^1} \sum_{\beta=1}^{n_{\ell(Z)}} \sum_{\Gamma \in \Gamma_{Z',even}^4} \widehat{G}_Z(\Gamma) T_{3,4}^i(\Gamma),$$

where $Z' = (Z, Z, z(\ell(Z), \beta), z(\ell(Z), \beta))$ and we have set

$$\widehat{G}_Z(\Gamma) \ = \ \prod_{i=\ell(Z)+1}^{d} \frac{2^{2-|\Gamma(i)|}}{n_{i-1}^2} \mu_4^{\mathbf{1}_{\{|\Gamma(i-1)|=|\Gamma(i)|=1\}}}.$$

Notice that $T_{3,4}^i = 1$ if $i \leq \ell(Z)$. Moreover, for $i \geq \ell(Z) + 1$, the same argument as in the proof of Lemma 8 shows that the number of $\Gamma \in \Gamma_{Z',even}^4$ for which $\gamma_3, \gamma_4$ pass through the same edge at layer $i$ and correspond to the same unordered multiset of edges $E$ equals

$$6^{\#\mathrm{loops}(E)-\mathbf{1}_{\{|R(E(i))| \cdot |L(E(i))| \neq 1\}}} \ \simeq \ 6^{\#\mathrm{loops}(E)}.$$

As in the proof of Proposition 3, observe that $\widehat{G}_Z(\Gamma) T_{3,4}^i(\Gamma)$ depends only on the unordered multiset of edges $E^\Gamma$ in $\Gamma$. Thus, we find that

$$\mathbb{E}[K_{\mathrm{b}}K_{\mathrm{w}}] \ \simeq \ \frac{d\|x\|^2}{n_0} \sum_{Z \in Z^1} n_{\ell(Z)} \sum_{E \in \Sigma_{Z',even}^4} G_Z(E) 6^{\#\mathrm{loops}(E)}.$$

Applying Proposition 10 as in the end of the proof of Propositions 3 and 4 we conclude

$$\sum_{E \in \Sigma_{Z',even}^4} G_Z(E) 6^{\#\mathrm{loops}(E)} \ = \ \frac{1}{n_{\ell(Z)}^2} \exp\left(5 \sum_{i=\ell(Z)+1}^{d} \frac{1}{n_i}\right) \left(1 + O\left(\sum_{i=1}^{d} \frac{1}{n_i^2}\right)\right).$$

Hence,

$$\mathbb{E}[K_{\mathrm{b}}K_{\mathrm{w}}] \ \simeq \ \frac{d\|x\|^2}{n_0} \left[\sum_{j=1}^{d} \exp\left(-5 \sum_{i=1}^{j} \frac{1}{n_i}\right)\right] \exp\left(5 \sum_{i=1}^{d} \frac{1}{n_i}\right) \left(1 + O\left(\sum_{i=1}^{d} \frac{1}{n_i^2}\right)\right).$$

To complete the proof of Proposition 5 it remains to evaluate $\mathbb{E}[\Delta_{\mathrm{wb}}]$. To do this, we note that, as in the proof of Lemma 6, we have

$$\Delta_{\mathrm{wb}} \ = \ \sum_{\substack{a \in [n_0]^2 \\ a=(a_1,a_2)}} \prod_{k=1}^{2} x_{a_k} \sum_{Z \in Z^1} \mathbf{1}_{\{y_Z > 0\}} \sum_{\Gamma \in \Gamma_{(Z,Z,a_1,a_2)}^4} \sum_{e \in \gamma_2, \gamma_3} \frac{\prod_{k=1}^{4} \mathrm{wt}(\gamma_k)}{\widehat{W}_e^2}$$

plus a term that has mean 0. Therefore, as in Lemma 7, we find

$$\mathbb{E}[\Delta_{\mathrm{wb}}] \ \simeq \ \frac{\|x\|_2^2}{n_0} \sum_{Z \in Z^1} \sum_{\Gamma \in \Gamma_{(Z,Z,1,1),even}^4} P(\Gamma) \#\{\text{edges } e \text{ belonging to both } \gamma_2, \gamma_3\},$$

where

$$P(\Gamma) \;=\; \prod_{i=1}^{\ell(Z)} \frac{1}{n_{i-1}} \mathbf{1}_{\left\{\substack{\gamma_3(i-1)=\gamma_4(i-1)\\ \gamma_3(i)=\gamma_4(i)}\right\}} \prod_{i=\ell(Z)+1}^{d} \frac{2^{2-|\Gamma(i)|}}{n_{i-1}^2} \mu_4^{\mathbf{1}_{\{|\Gamma(i-1)|=|\Gamma(i)|=1\}}}.$$

Thus, we have

$$\mathbb{E}[\Delta_{\mathrm{wb}}] \;\simeq\; \frac{\|x\|_2^2}{n_0} \sum_{i=1}^{d} \sum_{Z \in Z^1} \sum_{E \in \Sigma^4_{Z',even}} P(E) T_{2,3}^i(E) \# \left\{ \Gamma \in \Gamma^4_{Z',even} \Big|^{E^\Gamma=E\ \gamma_2(i)=\gamma_3(i)}_{\gamma_2(i-1)=\gamma_3(i-1)} \right\},$$

where $T_{2,3}^i(E)$ is as in (29), the sum is over unordered edge multisets $E$ (see (17)), and we've set

$$Z' = (Z, Z, z(0,1), z(0,1)).$$

As in Lemma 8, the counting term satisfies

$$\# \left\{ \Gamma \in \Gamma^4_{Z',even} \Big|^{E^\Gamma=E\ \gamma_2(i)=\gamma_3(i)}_{\gamma_2(i-1)=\gamma_3(i-1)} \right\} \;\simeq\; \mathbf{1}_{\{\ell(Z)<i\}} C(E, \ell(Z), i) 6^{\#\mathrm{loops}(E)},$$

where $C(E, i, j)$ was defined in (32) and is the event that there exists a collision between layers $i, j$ (i.e. there exists $\ell = i, \dots, j-1$ so that $|R(E(\ell))| = 1$). Proceeding now as in the derivation of $\mathbb{E}[\Delta_{\mathrm{ww}}]$ at the end of the proof of Proposition 3, we find

$$\mathbb{E}[\Delta_{\mathrm{wb}}] \;\simeq\; \frac{\|x\|_2^2}{n_0} \exp\left(5 \sum_{i=1}^{d} \frac{1}{n_i}\right) \left[ \sum_{\substack{i,j=1\\ j<i}}^{d} e^{-5 \sum_{\alpha=1}^{j} \frac{1}{n_\alpha}} \sum_{\ell=j}^{i-1} \frac{1}{n_\ell} e^{-6 \sum_{\alpha=j+1}^{\ell-1} \frac{1}{n_\alpha}} \right] \left(1 + O\left(\sum_{i=1}^{d} \frac{1}{n_i^2}\right)\right).$$

This completes the proof of Proposition 5. $\hspace{1em}\square$

