# OpenReview forum: "Finite Depth and Width Corrections to the Neural Tangent Kernel"
_ICLR.cc/2020/Conference — Accept (Spotlight)_

### Official Review · AnonReviewer3 · 2019-10-22
**Official Blind Review #3**

**Rating:** 8

**Review:**

This is an important contribution to understand finite depth and width corrections to the NTK. The authors show that the diagonal terms of NTK remain stochastic when depth and width approach infinity at the same rate.

NTK [1] is one of the most exciting discovers for extremely over-parameterized NNs in the last year. In the single limit setting, i.e. fixing depth and letting width -> infinity, the [1] showed that the NTK converges in distribution to a deterministic kernel and remained almost unchanged during gradient descent.  This regime is known as the kernel regime or linearized regime, where the training dynamics of the NN is well-approximated by its first order Taylor expansion.

In this paper, the authors show that in the double limit regime, i.e. depth/width = \beta and depth, width -> infinity, the diagonal terms of the NTK, as well as the first gradient step, is NOT deterministic. More precisely, they upper and lower bound the second moment of the diagonal terms of NTK (and first gradient step) through the temperature \beta. Their method builds on the `"sum-over-path approach" developed in [3], etc.

Overall, this is a very interesting result, proposing a new scaling limit that gradient descent dynamics can be highly nontrivial (i.e. not in the kernel regime.) and NNs can possibly learn useful representation.

Other comments:
1. It will be very helpful to have some experiments to support the main theorems in the paper since the proof is quite involved.
2. How difficult is it to compute the off-diagonals? Is it possible to obtain other statistics of of the NTK, trace, max eigenvalue, etc.
3. Is it possible to extend the results to other non-linearities, e.g. Tanh?


[1]Arthur Jacot, Franck Gabriel, and Clement Hongler. Neural tangent kernel: Convergence and gen- ´
eralization in neural networks. In Advances in neural information processing systems, pp. 8571–
8580, 2018.
[2] Jaehoon Lee, Lechao Xiao, Samuel S Schoenholz, Yasaman Bahri, Jascha Sohl-Dickstein, and Jeffrey Pennington. Wide neural networks of any depth evolve as linear models under gradient
descent. arXiv preprint arXiv:1902.06720, 2019.
[3]Boris Hanin and David Rolnick. How to start training: The effect of initialization and architecture.
In Advances in Neural Information Processing Systems, pp. 571–581, 2018

**Experience Assessment:**

I have published in this field for several years.

**Review Assessment: Checking Correctness Of Derivations And Theory:**

I assessed the sensibility of the derivations and theory.

**Review Assessment: Checking Correctness Of Experiments:**

N/A

**Review Assessment: Thoroughness In Paper Reading:**

I read the paper at least twice and used my best judgement in assessing the paper.

---

> ### Author Response · Authors · 2019-11-11
> **We thank the reviewer for his/her comments and address them in detail.**
>
> We thank the reviewer for his/her careful reading of our paper and various questions. We respond to them in order:
>
> 1. About experiments, please see point 1 in our response to Reviewer #1 and point 1c in our response to Reviewer #2. However, we would like to point out that in lieu of having simulations, we devote Section 3.1 to giving an informal  overview of the main steps and ideas in our proofs.
>
> 2. As explained in point 1b in our response to Reviewer #2, computing finite depth/width corrections to the statistics for the off-diagonal entries K(x,x') of the NTK appears to be challenging at the moment. We are certainly thinking about it and hope to address it in future work. We also think that computing spectral statistics of the NTK would be very interesting but may involve even more difficulties since it would may require knowing not only the joint distribution of the terms such as K(x,x') but also of all the entries in the full dataset by dataset size kernel matrix. Thank you for bringing up this point: we will add a short discussion in our revision.
>
> 3. As mentioned in point 1b of our response to Reviewer #2, we do not know at the moment if our techniques (and indeed the qualitative nature of our results) would extend to other non-linearities. Intuitively, bounded non-linearities may lead to very different spectral statistics for the NTK. For instance, even in the infinite width limit different non-linearies give rise to different spectral statistics for the much simpler computation of the input-output Jacobian. Again, we thank you for bringing up this point and will mention this issue in our revision.

---

### Official Review · AnonReviewer1 · 2019-10-22
**Official Blind Review #1**

**Rating:** 8

**Review:**

The paper investigates a novel infinite width limit taking depth to infinite at the same time. This is beyond conventional theoretical studies for infinite width networks where depth is kept finite when the width is taken to be infinite. The main object that paper studies is the neural tangent kernel which is of great interest to the theoretical deep learning community as it describes gradient descent dynamics in a tractable way.

While standard NTK becomes deterministic in the infinite width limit, when both depth and width are simultaneously taken to infinity this paper shows that NTK is no longer deterministic. Moreover authors show that gradient descent update induce non-negligible change to the Kernel.

There are two main limitations of otherwise significant work. One is generality of sums over path method used here beyond ReLU/Fully connected/single input setting. It is a very powerful technique allowing tight upper and lower bound for variation of diagonal entry of NTK. I worry that the method may be too specific to the particular network setting. Still this does not eclipse the strong results it could say for ReLU networks.

Second limitation is lack of empirical check. I understand it may be non-trivial to simulate double scaling limit and theoretical contribution alone could be significant progress. I still believe that empirical support should be a strong foundation of science of neural networks and this paper would improve even with some toy model implication of simultaneous depth/width limit. One might particularly wonder, for sufficiently deep/wide network that we could train on our computer, can we observe effects of d/n ( or \sum_i  1/n_i)?

Question:
I did not quite comprehend the alluded connection to double descent curve with data-dependent features in section 3.2. Could you elaborate?

nit : Dyer&Gur-Ari’s workshop paper is from the 2019 ICML workshop instead of 2018.
p3 sentence below eq (5) unnecessary ‘them’ in the end of line.


**Experience Assessment:**

I have published one or two papers in this area.

**Review Assessment: Checking Correctness Of Derivations And Theory:**

I assessed the sensibility of the derivations and theory.

**Review Assessment: Checking Correctness Of Experiments:**

N/A

**Review Assessment: Thoroughness In Paper Reading:**

I read the paper at least twice and used my best judgement in assessing the paper.

---

> ### Author Response · Authors · 2019-11-11
> **We thank the reviewer for his/her comments and respond to them in detail.**
>
> We thank the reviewer for his/her comments and feedback, which we address point-by-point below.
>
> 1. We agree that, at the moment, the sum-over-paths techniques we use to study the NTK in ReLU networks do not seem to readily extend to other architectures (e.g. convolutional networks, especially with pooling) and other non-linearities. We acknowledge this in the conclusion and will add another sentence or two describing the concomitant difficulties.
>
> 2. While we agree that ideally we would add experiments to our paper we have decided to not include such experiment in this paper. Please see point 1c of our response to Reviewer #2.
>
> 3. We agree that we should amplify our discussion in Section 3.2 of the relation of our work to the double descent curve and will do so in our revision. The essential point we wanted to get across is that there are many different modes of overparameterization (e.g. finite depth and infinite width or simultaneous infinite depth/width). Since in the deep and wide limit the kernel can in principle be data-adaptive, this raises the prospect that the second descent in the double descent curve may be deeper (better generalization) in this regime.
>
> 4. We will also update the reference to Gur-Ari and Dyer and will remove the unnecessary "them" below (5). Thank you for catching these mistakes.

---

### Official Review · AnonReviewer2 · 2019-10-23
**Official Blind Review #2**

**Rating:** 6

**Review:**

This paper studies the finite depth and width corrections to the neural tangent kernel (NTK) in fully-connected ReLU networks. It gives sharp upper and lower bounds on the variance of NTK(x, x), which reveals an exponential dependence on a quantity beta=d/n, where d is depth, and n is hidden width. This implies that when beta is bounded away from 0, NTK(x, x) is not deterministic at initialization. The paper further analyzes the change of NTK(x, x) after one step of SGD on a single datapoint x, and shows that the change also depends exponentially on beta.

NTK has been a popular subject of theoretical study in deep learning, and it's an important question to understand when and to what extent NTK can capture the behavior of real neural networks. This paper makes partial progress by analyzing the diagonal entries of the NTK in fully-connected ReLU networks. It concludes that NTK(x, x) at initialization is not deterministic, and can change significantly after doing one SGD step on x, when beta is bounded away from 0. While it's nice that the authors obtained precise answers to these two questions, there are some drawbacks that limit the significance of this paper:

1. The entire paper only considers one single datapoint x, so it doesn't apply to the non-diagonal elements NTK(x, x') or the usual SGD with mini-batches containing multiple datapoints. Of course, it's already implied by the current paper that the NTK is not deterministic and can move a lot when beta is large, but it's unclear whether the reverse is true, i.e., what is the regime when the NTK becomes deterministic and frozen when an entire dataset is involved. An answer (even empirically) to this question would make the paper much more complete.

2. The result is also not so surprising given [Hanin, 2018] (and possibly some other papers by the same author) which also obtained a similar exp(beta) dependence using a similar combinatorial approach.

3. Minor issues regarding incorrect or missing references:
-- "Further, kernel method-based theorems show that even in this infinitely overparameterized regime neural networks will have non-vacuous guarantees on generalization (Wei et al., 2018)." The result of Wei et al. is not about kernel method and in particular not NTK.
-- "In fact, empirically, networks with finite but large width trained with initially large learning rates often outperform NTK predictions at infinite width." The authors should refer to the work of [Arora et al., 2019] which AFAIK is the first paper that provides empirical study of the (convolutional) NTK predictor at infinite width on relatively large datasets like CIFAR-10.

[Hanin, 2018] Which Neural Net Architectures Give Rise to Exploding and Vanishing Gradients?
[Arora et al., 2019] On Exact Computation with an Infinitely Wide Neural Net


--------
update: Thanks to the authors for the detailed reply, which answers most of my questions. I am updating my rating to weak accept in light of this.

Regarding Wei et al.: Their result on NTK is negative, i.e., they aimed to show that NTK is inferior to regularized NN. I find the way you cited this paper a bit weird.

**Experience Assessment:**

I have published in this field for several years.

**Review Assessment: Checking Correctness Of Derivations And Theory:**

I assessed the sensibility of the derivations and theory.

**Review Assessment: Checking Correctness Of Experiments:**

N/A

**Review Assessment: Thoroughness In Paper Reading:**

I read the paper at least twice and used my best judgement in assessing the paper.

---

> ### Author Response · Authors · 2019-11-11
> **We thank the reviewer for his/her comments and provide a detailed reply.**
>
> We thank the reviewer for his/her comments regarding and address them in order below. We were disappointed by the reviewer's low marks, and we hope that in light of our response the reviewer will look more favorably on our submission.
>
> Point #1:
>
> 1a. We agree that studying the off-diagonal entries K(x,x') in the NTK would give a more complete picture of its behavior as a function of depth/width. However, this is certainly difficult  theoretically and, we believe, non-trivial empirically (see 1b,1c below).
>
> 1b.   On the theory side, K(x,x') involves the joint distribution of activations and gradients at x and x'. At infinite width such inter-layer correlations are probably obtainable, but finite width corrections appear to be challenging. We are unaware of any technique for estimating the joint distribution of activation patterns (which neurons are on/off) at x,x' that works simultaneously for large depths/widths. We hope to return to this in future work. We had not fully appreciated the relevance of both the empirical and theoretical results in [Arora et. al. 2019]. As the reviewer suggests, we will cite the experiments therein in our discussion of the efficacy of kernel methods and will point out the relevance of the theoretical results (specifically Theorems 3.1 and 3.2) in our prior work section.
>
> 1c.    On the simulation side, it seems to us that it is not a simple matter to simulate the behavior of the NTK in the very deep and wide regime we are considering. If our networks had depth and width 1000, for instance, they would have on the order of $10^6$ parameters. Moreover, since we are trying to simulate random variables that we believe have significant variance, we expect that any experiments would require sizeable computational resources. Such numerics are certainly possible and we would like to do them in the future. However, given we'd ideally like to run a suite of such experiments, we leave them to a future empirical paper with the belief that the purely theoretical results we have obtained are meaningful on their own.
>
> Point #2:
>
> The reviewer is correct that the method employed here has a similar flavor that of [Hanin 2018]  in that we start with an sum-over-paths formula for the network function and its derivatives. However, there are substantial differences between the computations we carry out and previous results.
>
> 2a. Unlike prior work computing derivatives and activations in a fixed layer, the NTK involves significant inter-layer effects that require a more complicated analysis. (This is why our work only computes the variance of the NTK, while previous work was able to compute higher moments in the relevant quantities.) For instance the $\beta/n$ suppression of $E[\Delta K(x,x)]/E[K(x,x)]$ in Theorem 2 results from a delicate correlation between paths and is new. Further, since the NTK is a sum over all layers of similar (but not identically distributed and not independent) random variables, it is a priori unclear whether the log-normal behavior of each of them (which could reasonably have been predicted from prior work) persists or if the sum would has Gaussian fluctuations as a cursory application of the CLT suggests.
>
>
> 2b. The joint impact of the intermediate layers has another non-trivial effect on the final results. Namely, the prefactors in Theorems 1, 2 reveal an intricate architecture-dependence that no prior work had predicted.
>
> 2c. The NTK also explicitly depends on the derivative of the network function w.r.t. network biases. Prior work on  the input-output Jacobian in finite width ReLU nets did not need treat these random variables. Computing the interaction from weight-bias and bias-bias terms (e.g. the expectation $E[(\partial N/\partial b) (\partial N/\partial W)]$ in Proposition 4) is therefore novel.
>
> In summary, we believe that the computations here substantially extend previous results about the exponential exp(beta) dependence and were not definitely not obvious (at least to us).
>
> Point #3:
>
> 3a. Perhaps we are misunderstanding the reviewer's comments regarding the results of [Wei et al. 2018]. In the abstract of that paper, its authors write:
>
> "we construct a simple distribution in d dimensions which the optimal regularized neural net learns with O(d) samples but the NTK requires Omega($d^2$) samples to learn"
>
> and
>
> "we develop a new technique for proving lower bounds for kernel methods, which relies on showing that the kernel cannot focus on informative features.''
>
> We agree that a major focus of [Wei et. al. 2018] is the salutary effect of regularization. But interesting results about generalization of NTK-type kernel methods are also given in Theorems 1.1, 2.1, which are the results we were referring to. We will add a few sentences of explanation regarding the relevance of this reference in our revision.
>
> 3b. We have added a reference to [Arora et al., 2019] to explain the case of convolution networks. Thank you for pointing out this relevant work.

---

### Decision · Program_Chairs · 2019-12-19

**Decision:**

Accept (Spotlight)

**Comment:**

This paper aims to study the mean and variance of the neural tangent kernel (NTK) in a randomly initialized ReLU network. The purpose is to understand the regime where the width and depth go to infinity together with a fixed ratio. The paper does not have a lot of numerical experiments to test the mathematical conclusions. In the discussion the reviewers concurred that the paper is interesting and has nice results but raised important points regarding the fact that only the diagonal elements are studied. This I think is the major limitation of this paper. Another issue raised was lack of experimental work validating the theory. Despite the limitations discussed above, overall I think this is an interesting and important area as it sheds light on how to move beyond the NTK regime. I also think studying this limit is very important to better understanding of neural network training. I recommend acceptance to ICLR.